# Neglected Anatomical Areas in Ovarian Cancer: Significance for Optimal Debulking Surgery

**DOI:** 10.3390/cancers16020285

**Published:** 2024-01-09

**Authors:** Stoyan Kostov, Ilker Selçuk, Rafał Watrowski, Svetla Dineva, Yavor Kornovski, Stanislav Slavchev, Yonka Ivanova, Angel Yordanov

**Affiliations:** 1Research Institute, Medical University Pleven, 5800 Pleven, Bulgaria; drstoqn.kostov@gmail.com; 2Department of Gynecology, Hospital “Saint Anna”, Medical University—“Prof. Dr. Paraskev Stoyanov”, 9002 Varna, Bulgaria; ykornovski@abv.bg (Y.K.); st_slavchev@abv.bg (S.S.);; 3Department of Gynecologic Oncology, Ankara Bilkent City Hospital, Maternity Hospital, 06800 Ankara, Turkey; ilkerselcukmd@hotmail.com; 4Department of Obstetrics and Gynecology, Helios Hospital Müllheim, 79379 Müllheim, Germany; rafal.watrowski@gmx.at; 5Faculty Associate, Medical Center, University of Freiburg, 79106 Freiburg, Germany; 6Diagnostic Imaging Department, Medical University of Sofia, 1431 Sofia, Bulgaria; svetladineva7@gmail.com; 7National Cardiology Hospital, 1309 Sofia, Bulgaria; 8Department of Gynecologic Oncology, Medical University Pleven, 5800 Pleven, Bulgaria

**Keywords:** advanced epithelial ovarian cancer, anatomy, neglected anatomical areas, upper abdomen, optimal cytoreduction, omental bursa

## Abstract

**Simple Summary:**

Ovarian cancer (OC), the most lethal gynecological malignancy, usually presents in advanced stages. Unlike other gynecological malignancies, advanced epithelial OC often spreads through peritoneal and lymphatic dissemination to the upper abdomen. Hence, OC necessitates complex surgical procedures usually involving the upper abdomen with the aim of achieving optimal cytoreduction without visible macroscopic disease. Omitting dissection of these particular areas can compromise complete cytoreduction. Neglected anatomical areas that may harbor tumor residues include the omental bursa; Morison’s pouch; the base of the round ligament of the liver and hepatic bridge; the splenic hilum; and suprarenal, retrocrural, cardiophrenic and inguinal lymph nodes. These areas are commonly involved and should be rigorously evaluated in every patient with advanced epithelial OC as they often preclude optimal cytoreduction. This article provides a meticulous anatomical description of neglected anatomical sites concealing possible residual disease during OC surgery and describes surgical steps essential for the dissection of these “neglected” areas.

**Abstract:**

Ovarian cancer (OC), the most lethal gynecological malignancy, usually presents in advanced stages. Characterized by peritoneal and lymphatic dissemination, OC necessitates a complex surgical approach usually involving the upper abdomen with the aim of achieving optimal cytoreduction without visible macroscopic disease (R0). Failures in optimal cytoreduction, essential for prognosis, often stem from overlooking anatomical neglected sites that harbor residual tumor. Concealed OC metastases may be found in anatomical locations such as the omental bursa; Morison’s pouch; the base of the round ligament and hepatic bridge; the splenic hilum; and suprarenal, retrocrural, cardiophrenic and inguinal lymph nodes. Hence, mastery of anatomy is crucial, given the necessity for maneuvers like liver mobilization, diaphragmatic peritonectomy and splenectomy, as well as dissection of suprarenal, celiac, and cardiophrenic lymph nodes in most cases. This article provides a meticulous anatomical description of neglected anatomical areas during OC surgery and describes surgical steps essential for the dissection of these “neglected” areas. This knowledge should equip clinicians with the tools needed for safe and complete cytoreduction in OC patients.

## 1. Introduction

Ovarian cancer (OC) is a rare disease with specific tumor biology and clinical behavior. Therefore, OC represents one of the major causes of lethality from cancer among women in developed countries [1]. The majority of patients with advanced epithelial ovarian cancer (AEOC) are initially diagnosed at an advanced stage of the disease [1,2]. The main routes of spread include peritoneal and lymphatic dissemination with the upper abdomen being commonly affected in advanced stages, which, in turn, increases the rate of lymph node and peritoneal metastatic involvement and decreases the chance for complete cytoreduction [1]. Therefore, the surgical approach to AEOC has changed in the last few decades [3,4,5]. Optimal cytoreduction with no macroscopic visible disease (RO) remains the most important prognostic factor [3,6,7]. The proficiency and anatomical expertise of surgical teams significantly influence the quality of optimal cytoreduction. Suboptimal cytoreduction often arises from the neglect of potential anatomical sites predisposed to concealing macroscopic tumor residues, often left unexplored during AEOC surgery [8]. Omitting dissection of these particular areas can compromise complete cytoreduction [9]. Anatomical sites that may harbor “neglected” tumor residues include the omental bursa; Morison’s pouch; the base of the round ligament of the liver and hepatic bridge; the splenic hilum; and suprarenal, retrocrural, cardiophrenic and inguinal lymph nodes [3,5,7]. A profound understanding of anatomy is a prerequisite since in most cases the surgeon has to perform steps like liver mobilization, diaphragmatic peritonectomy and splenectomy, as well as dissection of suprarenal, celiac and cardiophrenic lymph nodes [9]. Consequently, oncogynecologists are responsible for the safe, precise and complete dissection of these anatomical areas. Anatomical areas such as the retroperitoneal pelvic and paraaortic lymph nodes, diaphragmatic peritoneum, mesentery of small intestine/colon, gallbladder and omentum are not included in the article, as these areas are always preciously investigated in cases of abdominal exploration during OC surgery.

The aim of the present article is to describe the potentially neglected anatomical areas during surgery for AEOC. Additionally, most of the surgical maneuvers useful in dissecting these areas are described in detail. The neglected anatomical areas in AEOC are shown in Figure 1.

## 2. Omental Bursa 

### 2.1. Boundaries

The omental bursa (OB), also referred to as the lesser peritoneal sac, is a natural space situated between the stomach and the pancreas [10]. The boundaries of the OB are defined as follows [5,10,11,12]: 

Anterior: The hepatogastric ligament (pars flaccida), the posterior wall of the stomach, the gastrocolic ligament. 

Posterior: The parietal peritoneum covering the right crura of the diaphragm, the abdominal aorta, the celiac trunk, the pancreas, the left suprarenal gland and the medial part of the anterior aspect of the left kidney and the duodenum.

Superior: The narrow between the right side of the esophagus and the ligamentum venosum fissure. 

Inferior: The fusion line of the layers of the greater omentum and the transverse mesocolon.

Left lateral wall: Lower bound—the gastrosplenic ligament and the splenorenal ligament; left gastroomental fold; upper bound—the gastrophrenic ligament.

Right lateral wall: The epiploic foramen (Winslow’s foramen). 

The OB can also be divided into an infragastric and a supragastric part. The infragastric part is located posterior to the greater omentum, caudally and posterior to the stomach. Surgeons may encounter this part of the OB during supracolic (total) omentectomy. The supragastric part is located posterior to the lesser sac and cranial to the pancreas. Accessing this part is more intricate than accessing the infragastric one [13]. 

#### 2.1.1. Recesses and Vestibule 

Within the OB, three peritoneal pouches or recesses can be identified. The superior omental recess is located between the caudal liver lobe and the diaphragm, whereas the inferior omental recess extends between the posterior wall of the stomach, the pancreas and the transverse mesocolon. More caudally, the inferior recess almost vanishes due to the fusion of the layers of the greater omentum. The superior recess communicates with the peritoneal cavity through Winslow’s foramen. The splenic recess is situated between the stomach and the hilum of the spleen [5,10,14].

The vestibule of the OB is located to the left of the epiploic foramen. It is bounded anteriorly by the hepatoduodenal ligament, superiorly by the caudate lobe of the liver and postero-inferiorly by the head of the pancreas [11,15]. 

#### 2.1.2. Hepatoduodenal Ligament and Foramen of Winslow

The greater omentum consists of the gastrosplenic, splenorenal, gastrocolic and gastrophrenic ligaments, whereas the lesser omentum is composed of the hepatogastric ligament and the hepatoduodenal ligament (HDL) [11]. The latter is of great interest, as the portal triad (common bile duct, proper hepatic artery, portal vein) is located beneath the two peritoneal leaves of the lesser omentum (visceral and parietal peritoneum)**.** The HDL forms a thick right-sided margin of the lesser omentum, connecting the porta hepatis of the liver and the superior duodenal flexure [11,15,16]. Between the two leaves of the HDL, the common bile duct runs right to the portal vein. The proper hepatic artery runs left to the portal vein [16]. The common bile duct and the proper hepatic artery are located anterior to the portal vein. The HDL also contains nerves, lymphatics, and fatty and connective tissue. The anterior vagal trunk of the vagus nerve is also a part of this complex ligamentous structure, and the lesser curvature of the stomach lies at the left part of the HDL, in close proximity with the anterior vagal nerve [11,15]. 

The foramen of Winslow (also referred to as the omental or epiploic foramen) is located posterior to the HDL. As mentioned above, this foramen is the only natural connection between the OB and the greater sac. The foramen has the following boundaries: anterior—the HDL; posterior—the parietal peritoneum covering the inferior vena cava, right crus of the diaphragm; inferior—the superior part of the duodenum; superior—the caudate lobe of the liver [11,12].

The anatomy of the supragastric OB is shown in Figure 2.

#### 2.1.3. Vessels 

The celiac trunk (CT), also referred to as the celiac axis, is the first visceral anterior branch of the abdominal aorta. It arises immediately after the aortic hiatus at the level of the T12/L1 vertebral bodies. The CT is approximately 1.5–2 cm long. It runs horizontally and above the splenic vein before trifurcating into the left gastric artery, splenic artery and common hepatic artery. This trifurcation is referred to as the “true” tripod because all three arteries share a common origin. When one of these arteries originates before the other two along the course of the CT, it is termed a “false” tripod. The left gastric artery is the smallest branch of the CT and lies slightly cranial to the remaining two arteries [11,15,17]. It passes between the two leaves of the lesser omentum to run along the lesser curvature of the stomach. The splenic artery, the largest branch, is slightly to the left of the common hepatic artery. The splenic artery is a tortuous branch and follows a leftward course slightly above the neck and tail of the pancreas. At the level of the neck of the pancreas, the artery runs horizontally before ascending and turning more laterally to terminate in the hilum of the spleen. The splenic artery gives off branches such as the left gastroepiploic artery and short gastric arteries. The common hepatic artery runs on the superior part of the duodenum. It divides into the gastroduodenal, proper hepatic and right gastric arteries [11,15,18]. The gastroduodenal artery is the first branch that runs caudally and supplies the pylorus, pancreas and duodenum. The right gastric artery follows a caudal course and passes within the two leaves of the lesser omentum along the lesser curvature of the stomach. The proper hepatic artery arises just after the origin of the gastroduodenal and right gastric arteries. It runs cranially and becomes a part of the portal triad between the two leaves of the HDL. The proper hepatic artery divides into the left and right hepatic arteries at the level of the porta hepatis [11,15,18].

The portal vein is the main vessel entering the liver, responsible for carrying about 75% of the blood flow. It arises from the confluence of the superior mesenteric vein and the splenic vein. The true origin of the portal vein begins immediately after the splenic–mesenteric confluence, which is located anterior to the IVC and posterior to the neck of the pancreas at the level of the second lumbar vertebra [19,20]. Three drainage patterns of the inferior mesenteric vein have been identified: into the splenic vein (type 1a), the superior mesenteric vein (type 1b) or the confluence of superior mesenteric and splenic vein (type 2) [21]. Notably, in the majority of cases, the inferior mesenteric vein and the left gastric vein drain into the splenic vein [11,15]. The portal vein enters the HDL and divides into left and right branches at the level of the porta hepatis [11,15].

#### 2.1.4. Porta Hepatis

The porta hepatis (PH) is a transverse nonperitoneal fissure located on the inferior surface of the liver from the gallbladder neck to the fissure for the ligamentum teres hepatis and ligamentum venosum. The PH is also delimited by the quadrate lobe in front and from the caudate process at the back. The lesser omentum connects to the PH margins. Moving from posterior to anterior, the left and right portal veins and the left and right hepatic arteries enter the PH. Conversely, some lymph nodes emerge from PH along with the left and right hepatic ducts [11,15,22].

The vessels of the OB are shown in Figure 3.

#### 2.1.5. Lymph Nodes 

Celiac lymph nodes are situated near the origin of the CT. These nodes are terminal, as they collect lymph from nodes located near the common hepatic, splenic and left gastric vessels. Celiac lymph nodes also drain lymph from most internal organs (liver, gallbladder, stomach, spleen and pancreas) into the cisterna chyli. Right and left small intestinal lymph nodes originate from the celiac nodes and form the small intestinal lymphatic trunk [5,11,15,23,24,25]. The number of celiac nodes varies from 3 to 15 [24,25].

The number of hepatic lymph nodes is variable. They can be divided into hepatic nodes (receiving lymph from the celiac nodes and located near the hepatic artery), subpyloric nodes (four or five nodes near the gastroduodenal artery) and cystic nodes (located at the neck of the gallbladder) [24,25]. The hepatic lymph nodes can be identified in both the PH and HDL [11,15]. The drainage of the hepatic lymph nodes can be classified into superficial and deep lymphatic networks. The superficial is later separated into three main groups, the most common being that passing through the HDL and gastrohepatic ligament. The deep pathway drains the lymph nodes at the liver hilum, and then from the hepatic lymph nodes to the nodes at the HDL. The latter can be divided into two chains—the posterior periportal chain and the hepatic artery chain. The hepatic chain drains into the celiac lymph nodes and then into the cisterna chyli [23,25].

### 2.2. Omental Bursa and Ovarian Cancer

The spread of OC into the OB occurs primarily by two routes—transcoelomic (peritoneal) spread or progressive lymph node involvement [5].

#### 2.2.1. Transcoelomic Metastases

There are mainly two hypotheses that have been described for the transcoelomic metastasis model in OC. The “seed and soil” theory explains that tumor cells detach from the primary tumor and circulate within the peritoneal cavity through peritoneal fluid before seeding intraperitoneally. The peritoneal fluid and OC cells flow in particular directions in a clockwise rotation—influenced by gravity, they tend to accumulate in the most dependent sites. Subsequently, the intraperitoneal fluid follows a cephalad direction towards the upper abdomen due to the movement of the diaphragm and peristalsis of the bowels. However, anatomical limitations restrict their movement within certain parts of the peritoneal cavity. On the right side, peritoneal fluid passes from the pelvis through the right paracolic gutter, Morison’s pouch, and the OB via the foramen of Winslow. Intraperitoneal fluid flow also reaches the right subphrenic space, including the liver capsule and the diaphragm. However, the falciform ligament limits the flow from the right to the left subphrenic space. Conversely, on the left side, peritoneal fluid is confined by the phrenicocolic ligament to the left paracolic gutter at the level of the inframesocolic recces [26,27,28].

The “metaplasia theory” postulates that metastatic omental sites in OC are not true metastases, but rather a synchronous malignant transformation due to the common lineage between omentum and ovarian epithelium [26,28].

However, both theories are insufficient to fully explain the pathogenesis of peritoneal metastases in OC. The “seed and soil” theory does not explain the different distribution patterns of peritoneal carcinomatosis (some patients have more peritoneal disease in the upper abdomen than in the pouch of Douglas), whereas the “metaplasia” theory implies that ovarian peritoneal carcinomatosis spreads randomly in the abdominal cavity [26,28]. 

In AEOC, the OB is often affected via the transcoelomic route. Peritoneal metastases in the lesser sac can be found in the following anatomical structures: HDL, PH, medial aspect of Winslow’s foramen, caudate lobe of the liver, parietal peritoneum covering the posterior border of the lesser sac, fissure for ligamentum venosum, subpyloric space, peritoneum over the transverse mesocolon, and posterior surface of the hepatogastric and gastrocolic ligaments [5,13]. The subpyloric space is a cul-de-sac, located below the pylorus. Due to gravity, ovarian tumor cells accumulate in this space along with the peritoneal fluid or ascites [29].

As previously mentioned, the peritoneal spread of OC to the lesser sac is possible only through the foramen of Winslow, which is a connection between the lesser sac and the peritoneal cavity. Thus, transcoelomic lesser sac metastases are absent in cases of adhesions and occlusion of the epiploic foramen (e.g., as a result of previous surgeries in the upper abdomen, with cholecystectomy being the most common cause of adhesions and obliteration of the foramen). Transcoelomic spread to the OB is also linked with conditions such as ascites, peritoneal carcinomatosis, high peritoneal cancer index (PCI), involvement of Morison’s pouch and diaphragmatic dissemination [5]. However, tumor spread into the lesser sac does not consistently follow expected patterns. Supragastric lesser sac metastases are observed in 70% of patients with a normal supracolic omentum. Interestingly, the lesser omentum can remain unaffected in about one-fifth of patients with lesser sac metastases [13]. These findings show that there is no fully reliable predictor for the transcoelomic spread of OC to the OB. Hence, a thorough assessment of the lesser sac is imperative in every patient undergoing cytoreductive surgery for AEOC. If upper abdominal disease is detected, the OB should always be opened and checked. Moreover, the surgeon must be careful of the adhesions because they may contain metastatic nodules. 

The percentage of lesser sac peritoneal carcinomatosis has been estimated in a few reports; the majority of studies combine descriptions of both peritoneal and lymph metastases [5,13,29]. Mukhopadhyay et al. reported lesser sac peritoneal metastases in 64% of patients with AEOC. Celiac lymph node metastases were excluded from the study. A PCI equal to or greater than 17 and involvement of Morison’s pouch were identified as the strongest multivariate predictors for lesser sac involvement [13]. Raspagliesi et al. found that 67% of women with AEOC had OB involvement, either peritoneal or lymphatic [5]. The authors specifically estimated transcoelomic dissemination to the OB in 59% of patients, with 81% having supragastric lesser sac involvement and 19% having peritoneal dissemination at the HDL [5]. Tozzi et al. investigated the dissemination of PH and hepato-celiac lymph nodes in 216 patients with AOC. Among these, 31 patients (14.3%) had a tumor on both anatomical sides, and out of these, 18 (8.3%) patients had only HDL involvement [30]. 

Transcoelomic tumor dissemination of the OB is shown in Figure 4.

#### 2.2.2. Omental Bursa Lymph Node Metastases

The other pathway of OC dissemination into the lesser sac is through the lymphatics. The following lymph nodes could be metastatic: triad (hepatoduodenal), portal and celiac lymph nodes (CLNs). The real incidence of CLN involvement is unclear as systematic lymph node dissection is not routinely performed in this region [5,7,30,31,32,33]. Studies have shown that patients with AOC only benefit from the removal of bulky nodes as part of optimal cytoreduction [34]. Therefore, the majority of studies included the rate of metastases among patients with suspicious CLNs [7,30,31,32,33]. Angeles et al. reported on 150 patients with AOC who underwent optimal cytoreduction. Seventeen (11.3%) women had CLN metastases [7]. Raspagliesi et al. reported on 3 patients (8%) with bulky metastatic CLNs among 37 women with AOC [5]. Gallota et al. observed metastases to the hepato-celiac lymph nodes in 52.9% of 85 patients who underwent hepato-celiac lymph node dissection. However, in their study, the hepato-celiac lymph nodes included also the portal and celiac triad lymph nodes [35]. Martinez et al. dissected CLNs in 41 women and found CLN metastases in 23 (56.1%) of the patients. However, the estimated percentage in this study could not be accurate as the authors included women with recurrent disease [32]. Patients with metastatic CLNs have higher PCI and more frequent PH involvement, as well as more frequent involvement of the mesenteric and paraaortic lymph nodes [5,7,32,33,35]. Martinez et al. reported that 81.1% of patients with CLN involvement had metastatic paraaortic lymph nodes [33]. Similarly, Angeles et al. observed that all patients with CLN metastases had metastatic disease of the paraaortic lymph nodes. The authors additionally found that more than 80% of patients with hepatic and lung recurrence had CLN involvement [7]. Martinez et al. reported that metastases to the CLNs are associated with extensive upper abdominal disease, hepatic metastases and a median PCI of 21. The authors also found that 20% of patients with CLN involvement had suspicious mediastinal lymph nodes on imaging tests (CT and PET-CT) [33]. Therefore, CLN dissection should be performed after a thorough preoperative evaluation of the mediastinal lymph nodes. The prognostic impact of CLN involvement is unfavorable as it is associated with decreased disease-free survival (DFS) and reduced overall survival (OS) due to short-term recurrences, increased risk of lymph node progression and resistance to platinum-based chemotherapy [7,32,33]. Furthermore, it is important to note that patients with CLN involvement experience poor outcomes even after undergoing optimal cytoreduction [7,32]. These findings prompt us to question the appropriateness of assigning these OC cases to FIGO stage IIIC [7,33,36,37]. In fact, there is a growing suggestion that this stage should be re-evaluated and possibly divided based on a distinct consideration of metastases to infrarenal lymph nodes and CLNs [7,33]. Notably, some experts have taken this notion a step further, advocating for classifying CLN metastases in line with FIGO stage IVB, a classification similar to that of patients with cardiophrenic lymph node involvement [7].

Comparably to the metastatic CLNs, the rate of metastases to the portal and triad lymph nodes in AOC patients is also hard to estimate. Donato et al. reported a rate of 4.5% for portal node metastases among 55 patients with AEOC and hepatobiliary involvement [31]. Song et al. identified portal lymph node involvement in 1.9% of patients undergoing primary cytoreduction for OC. However, recurrence rates in these nodes were notably as high as 16.7% [38]. Tozzi et al. observed hepato-celiac lymph node metastases in 16.1% of patients with AEOC and macroscopic disease at the PH and found PH and hepato-celiac lymph node involvement in approximately 15% of studied AOC cases [30]. In cases of paraaortic and mesenteric metastatic lymph nodes, it is essential to assess portal and triad lymph nodes during surgery [35]. The presence of portal or triad lymph node involvement is associated with poorer prognosis compared to uninvolved nodes. Involvement of lymph nodes in these regions serves as an indicator of disease severity, decreased DFS and reduced OS [31,35,39]. Retrospective data indicate that hepato-celiac lymph node metastases independently predict decreased progression-free survival (PFS) [35].

The sensitivity and specificity of different imaging modalities for detecting metastatic lymph nodes in the OB vary in the medical literature. One study reported good sensitivity (77%) but low specificity in detecting CLN metastases on CT, highlighting the importance of radiologist expertise (with nonexpert sensitivity at 20%) [7]. Another study indicated that pre-operative CT missed detecting most cases of PH and CLN metastases [5]. Retrospective data suggested that positron emission tomography (PET) CT scans are more sensitive than CT for detecting hepato-celiac lymph nodes and PH peritoneal dissemination [31]. However, another retrospective study reported low sensitivity in the detection of CLN metastases using preoperative PET-CT and CT scans [33]. Nevertheless, a different retrospective study found that a combination of preoperative CT and diagnostic laparoscopy detected all cases of hepato-celiac lymph node metastases and PH peritoneal involvement, with CT alone missing the disease in these particular regions in 31% of cases [30]. Nowadays, oncogynecologists deal with tumor peritoneal implants or lymph node dissemination in the omental bursa. An exception is transcoelomic or lymph node dissemination of the porta hepatis, where an interdisciplinary surgical approach with biliary surgeons is required [30,31,38].

CLN metastases are shown in Figure 5.

### 2.3. Surgical Approaches to the Omental Bursa

#### 2.3.1. Dissection of the Hepatogastric Ligament (Pars Flaccida)

To gain access to the supragastric part of the OB, the stomach should be retracted to the left, exposing the superior part of the pancreas. The hepatogastric ligament lies between the visceral surface of the left liver lobe and the lesser curvature of the stomach. The left section of the gastrohepatic ligament is thinner than other parts of the ligament because there is almost no fatty tissue between the peritoneal layers. It is also referred to as pars flaccida of the lesser omentum. This is the preferred anatomical entry point to the OB [12,15]. This approach enables access to the supragastric part of the OB. It is important to note potential anatomical variations of the celiac trunk. Particular attention is warranted for cases of a left hepatic artery arising from the left gastric artery (incidence 12–34%), and in rare instances, a common hepatic artery originating from the left gastric artery. In such scenarios, the anomalous hepatic artery crosses the supragastric part of the OB through the midline [40].

#### 2.3.2. Dissection of the Gastrocolic Ligament 

The gastrocolic ligament extends from the inferior two-thirds of the greater curvature of the stomach to the transverse mesocolon. On the left, it continues as the gastrosplenic ligament, whereas on the right, it is limited by the gastroduodenal junction. Four layers of the peritoneum that enclose the stomach and the transverse mesocolon/colon are part of the greater omentum. The layers that descend to form the greater omentum later fuse to become the two layers of the gastrocolic ligament at the level of the transverse mesocolon. The anterior layer of the gastrocolic ligament attaches to the greater curvature of the stomach, and the posterior layer attaches to the transverse mesocolon. In adults, the two layers on the right side of the ligament are in close proximity to each other and the transverse mesocolon. On the left side, there is a distance between the two layers of the ligament. Therefore, the left side of the gastrocolic ligament is the preferable point of dissection and entry to the OB [10,11,12,15,41].

While these two approaches are frequently employed in OC surgery, additional techniques will also be discussed. It is important for oncogynecologists to possess a basic familiarity with various surgical accesses to the OB.

#### 2.3.3. Anterior Trans-Omentum Approach

This approach represents a direct transection of the greater omentum at the level of the greater curvature of the stomach [42]. It is suitable for patients with a normal body mass index and a greater omentum characterized by minimal adipose tissue. However, injury to the gastroepiploic vessels can potentially compromise the dissection.

#### 2.3.4. Dissection of the Gastrosplenic Ligament

The gastrosplenic ligament forms through the lateral fusion of the peritoneal layers of the greater omentum. It is a thin attachment between the left part of the great stomach curvature and the hilum of the spleen. During dissection of the gastrosplenic ligament, the surgeons should be aware of the short gastric vessels and the left gastroepiploic vessels [10,11,12,15].

#### 2.3.5. Trans-Mesocolic Dissection 

In this technique, the transverse mesocolon is dissected above the inferior border of the pancreas. Surgeons should be aware of the possible presence of the Moskowitz artery, an anatomical variation found in up to 17% of patients. This artery, also known as the meandering mesenteric artery, forms a collateral pathway between the left colic and middle colic arteries, passing above the inferior border of the pancreas [42,43,44].

The different surgical approaches are shown in Figure 6.

#### 2.3.6. Kocher Maneuver

Dissection of the gastrocolic and gastrohepatic ligaments provides good access to the OB and celiac trunk but is insufficient for dissection of the HDL and PH. To fully expose these structures, a maneuver for the mobilization of the duodenum and head of the pancreas was first described by Theodor Kocher. This approach to duodeno-pancreatic mobilization, commonly used in visceral surgeries such as the Whipple procedure, or in emergent surgeries for retroperitoneal hemorrhage, and can also be beneficial in gynecologic-oncological procedures in cases of tumor dissemination involving the PH, HDL, and suprarenal lymph nodes. The Kocher maneuver starts with the medialization of the first, second and proximal third portions of the duodenum. A vertical incision of the parietal peritoneum is made 1–2 cm lateral to the second part of the duodenum. The incision extends perpendicularly between the lateral aspect of the epiploic foramen and the inferior duodenal flexure. The procedure continues with a gentle dissection of the fascia of Toldt (the peritoneal adhesion plane between the visceral peritoneum of the ascending mesocolon and the retroperitoneum), which is located lateral to the duodenum and the head of the pancreas. A further avascular plane containing loose connective tissue and allowing for easy and bloodless dissection is situated below the duodenum and the head of the pancreas and corresponds to the fusion fascia of Treitz (the adhesion plane between the visceral peritoneum of the duodenum and pancreas and the retroperitoneum). Both structures are covered from above by the visceral peritoneum and the fusion fascia of Fredet (the plane between the ascending mesocolon and the visceral duodenal–pancreatic peritoneum) [45,46,47,48,49]. 

The Kocher maneuver allows access to the infrahepatic IVC, duodenum, abdominal aorta, superior mesenteric artery, posterior surface of the head of the pancreas, right renal hilum and HDL. The limit of the dissection is the medial aspect of the IVC, determined by identifying the left renal vein. The inferior mesenteric vein has also been described as the medial limit. The dissection of the peritoneum can be carried out cranially up to the retrohepatic IVC, thereby enabling the dissection of the posterior part of the PH [5,30,46,47,48]. The precise incision point for entry is critical, as a more lateral incision could open the renal fascia and lose the right plane of dissection. Conversely, injury to the duodenum and vessels is possible if the incision is made more medially than usual [47]. The Kocher maneuver is often combined with the Cattel—Braasch maneuver, which represents a mobilization of the ascending colon from the retroperitoneum after dissection of Toldt’s fascia [46]. Currently, the Kocher maneuver is also performed by oncogynecologists [47]. The Kocher maneuver and fascias near the duodenum and pancreas head are depicted in Figure 7, Figure 8 and Figure 9.

#### 2.3.7. Dissection of Portal, Celiac and Triad Lymph Nodes

The full exposure of the OB is achieved through the dissection of the gastrocolic and gastrohepatic ligaments and the Kocher maneuver. The HDL is isolated using a vessel loop through the foramen of Winslow, enabling traction of the ligament. For hepatic resections, this vessel loop can additionally control the vascular flow of the liver (Pringle maneuver); the loop can close the hepatoduodenal vascular flow for up to 25–30 min [18,30]. Dissection begins with peritonectomy of the HDL within a tumor-free zone. The hepatic artery and common bile duct are dissected. The artery is identified after cranial retraction of the stomach and gentle caudal retraction of the pancreas. The peritoneum is dissected between the superior part of the duodenum and the ligamentum teres hepatis. The HDL is retracted medially with the vessel loop, and the posterior peritoneum of the ligament is dissected. The portal vein is identified, and all structures of the portal triad are visualized and mobilized. Enlarged lymph nodes at the PH, proper hepatic artery and common bile duct are meticulously separated and dissected [30,33,35]. Suspicious lymph nodes between the portal vein and infrahepatic IVC are removed after gentle medial traction of the portal vein with the vessel loop. In cases of other enlarged lymph nodes, the dissection proceeds in a retrograde manner along the gastroduodenal artery, right gastric artery and common hepatic arteries. This pathway leads to the celiac trunk, where enlarged lymph nodes are also resected. There are various dissection techniques; however, all authors start dissection immediately after the identification of all anatomical structures within the OB, which will prevent inadvertent injuries and enable immediate actions for bleeding complications. Starting lymphadenectomy from the arteries and using them as landmarks during dissection is a commonly employed approach in oncogynecology [30,33,35]. Some surgeons perform cholecystectomy for better exposure of the right side of the HDL and PH [33]. It should be stressed that in approximately 3.5% of cases, the common hepatic artery may originate from the superior mesenteric artery [50].

## 3. Morison’s Pouch

Morison’s pouch, also referred to as the posterior right subhepatic space or hepatorenal pouch, is part of the right supramesocolic compartment. It is separated from the anterior right subhepatic space by the transverse mesocolon. The posterior right subhepatic space communicates superiorly with the right subphrenic space, inferiorly with the right paracolic space, and medially with the lesser sac through Winslow’s foramen [51,52,53]. Morison’s pouch is defined by the following boundaries [51,52,53]:

Cranial: The posterior layer of the coronary ligament of the right liver lobe.

Caudal: The superior part of the right kidney.

Lateral: The parietal peritoneum, where it merges with that of the right diaphragm

Medial (from lateral to medial): Hepatic flexure of the transverse colon, transverse mesocolon, the second part of the duodenum and the PH.

Anterior: The right lobe of the liver and the gallbladder.

Posterior: The upper portion of the right kidney, right adrenal gland and infrahepatic IVC.

In normal conditions, Morison’s pouch is free of any fluid collection. However, fluid accumulation occurs in the pouch due to gravity, as it is one of the posterior spaces in the peritoneal cavity. For this reason, Morison’s pouch is one of the most affected anatomical zones in AEOC [9,53,54,55]. It is also a frequent site for OC recurrences [55]. However, the percentage of Morison’s pouch metastases in AEOC has not been estimated in the literature, as it is actually an upper abdomen metastatic peritoneal involvement. Morison’s pouch is routinely dissected by oncogynecologists [13].

The dissection begins with mobilization of the right liver lobe, involving the division of the anterior leaf of the coronary and triangular ligaments of the right liver lobe. The liver is retracted cranially following the transection of the posterior leaf of the coronary ligament. The transverse mesocolon is retracted caudally, and the duodenum is retracted medially. Dissection and resection of the right colic flexure (hepatic flexure) enable better access. The peritoneum is stripped from lateral to medial until the borders of the pouch are reached [53,55]. Special attention must be paid to avoid a potential injury to the right adrenal gland, infrahepatic IVC and right diaphragm [53]. Additionally, it must be noted that there are small veins, infra- and retrohepatic veins, draining directly into the IVC (Figure 10) [3].

## 4. Base of the Round Ligament and the Hepatic Bridge

The hepatic bridge, also known as pons hepatis or pont hepatique, is an anatomical variation in which the liver parenchyma bridges between segments III and IVb of the liver. Consequently, a tunnel forms over the umbilical fissure at the base of the round ligament [56,57,58,59]. Different pons hepatis types have been described. Initially, Couinaud classified it into three types: Type I: no communication, Type II: membranous communication, and Type III: a massive bridge [60,61]. Sugarbaker established a classification into four types: Type O—no hepatic bridge, where the base of the round ligament is visible; Type I—less than one-third of the umbilical fissure is covered by liver parenchyma; Type II—the hepatic bridge covers up to two-thirds of the fissure; Type III—more than two-thirds is covered by the liver parenchyma (Figure 11) [56]. Cawich et al. proposed a dichotomous typology of pons hepatis—incomplete (the liver fissure being incompletely covered by the liver parenchyma, <2 cm) and complete (the fissure is completely covered, >2 cm) [62]. The frequency of the hepatic bridge varies in the literature. Sugarbaker found a frequency of 49% in 102 patients [56], while Cawich et al. reported a prevalence of 40.9% in 66 cadavers [62]. Other studies reported lower incidences—22.85% and 30%, respectively [63,64]. 

The spread of peritoneal tumors, including OC, can invade the walls of the tunnel. In such cases, the hepatic bridge should be resected to assess the tunnel. The possibility of tumor implants in the tunnel is high when there is macroscopic peritoneal disease at the base of the round ligament, as it is a continuation of peritoneal tissue [58]. Surgeons should remain vigilant about the base of the round ligament even without the presence of a hepatic bridge, as it can harbor macroscopic tumor tissue (Figure 12). Gulmez et al. investigated 101 patients with peritoneal carcinomatosis who underwent cytoreductive surgery and hyperthermic intraperitoneal chemotherapy. Patients were diagnosed with mucinous adenocarcinoma of the appendix, malignant peritoneal mesothelioma, and colorectal and ovarian cancers. The authors found tumor implants in 18 patients (28.6%) among 63 who underwent distal round ligament resection. The study concluded that in OC patients, the round ligament should be resected in cases of PCI ≥ 10 [59]. However, the absence of peritoneal metastases at the Glisson capsule or PH can be an indication that the tunnel is not affected by the tumor [56]. The frequency of OC spread to the distal round ligament and hepatic bridge is not well reported. 

The presence of peritoneal implants on the round ligament of the liver is an indication for ligament resection down to its entrance into the liver parenchyma. Transection of the round ligament of the liver starts with resection of the falciform ligament close to the anterior abdominal wall. Cranial traction of the falciform ligament provides control and precise peritoneal dissection at the base of the round ligament of the liver. The base of the ligament is clamped, resected and sutured with absorbable stitches. The suture is used to prevent bleeding from a patent umbilical vein (a patent connection between the fetal umbilical vein and the portal system). From a practical point of view, it should be also noted that there is a risk of injury of the left portal vein during resection at the base of the round ligament [56,57,58]. 

Dissection of the hepatic bridge could be associated with injury of vital structures, which pass into the left lobe of the liver. The left hepatic duct and left hepatic artery are at risk of iatrogenic injury during resection of the hepatic bridge. Therefore, just after opening the posterior and anterior aspects of the hepatic tunnel, it is advisable to use a vessel loop that passes through the hepatic bridge. This maneuver will enable a safe and bloodless dissection [56]. An interdisciplinary team approach with biliary surgeons is required for the transection of the base of the round ligament of the liver and the hepatic bridge, as an oncogynecologist will hardly manage intraoperative and especially postoperative complications (biliary leak) [56,59].

## 5. Hilum of the Spleen

Three types of metastatic patterns of splenic involvement in AEOC have been proposed: hilar, capsular and parenchymal patterns [65,66]. Transcoelomic involvement of the splenic hilum is possible by two routes, via the OB and the greater omentum. Ovarian tumor cells spread to the OB and then reach the splenic hilum through the posterior part of the gastrosplenic ligament [5]. The splenic hilum is also affected in the case of massive metastatic omentum majus involvement (omental cake). The greater omentum is attached to the spleen, and the splenorenal ligament is considered part of the omentum. Moreover, as mentioned above, the greater omentum is continuous on the left side with the gastro-splenic ligament (Figure 13) [11,15]. Sugarbaker emphasized that some types of splenic parenchymal metastases should be considered as peritoneal dissemination rather than hematogenous metastases. He reported that splenectomy might be beneficial for AEOC patients with peritoneal carcinomatosis and no other distant parenchymal metastases. In such cases, splenic parenchymal metastases are actually peritoneal, and splenectomy is part of optimal cytoreduction [66]. This theory is supported by the fact that parenchymal spleen involvement is also observed in patients with pseudomyxoma peritonei, a disease with progressive dissemination through the peritoneal cavity [66]. The incidence of hilar splenic metastases varies in the literature. However, the majority of authors reported a rate equal to or more than 50% in patients with AEOC, and hilar metastases are more common than parenchymal metastases [65,67,68,69,70]. Women with splenic hilar metastases have a significantly shorter survival time compared to patients without hilar disease. In addition, women with parenchymal splenic involvement have a decreased OS rate compared to women with hilar metastases [65]. Therefore, assessment of the splenic hilum remains mandatory in all patients with AEOC. It should be remembered that involvement of the OB and involvement of the greater omentum are predictors of splenic hilar metastases. Splenic flexure and cutting the peritoneum posterior to the spleen elevate the spleen from the deep fossa and provide a detailed exposure. This approach also improves the dissection of the greater omentum at the splenic level. Currently, oncogynecologists often perform splenectomy. However, the incidence of injury to the tail of the pancreas is not rare. In such cases, interdisciplinary discussion on postoperative management could be required (biliary surgeons) due to an eventual pancreatic ductal leak or fistula [68,70].

### Extraperitoneal Lymph Node Zones and Lymphadenectomy

The extraperitoneal lymph node zones aside from the pelvic and infrarenal lumboaortic zone are suprarenal, retrocrural, cardiophrenic and inguinal. The lymphadenectomy in ovarian neoplasms (LION) trial showed that patients (group 1) with normal lymph nodes (before and during surgery) who underwent infrarenal paraaortic lymph node dissection were not associated with increased overall or progression-free survival compared to patients in the no-lymphadenectomy group (group 2). Additionally, patients in group 1 had an increased incidence of postoperative complications compared to patients in group 2 [34]. This study gives insights into the role of lymphadenectomy in normal lymph nodes. Since then, selective lymph node dissection in AEOC (irrespective of lymph node locations) has only been performed in cases of suspected lymph nodes (enlarged and debulked lymph nodes) as part of optimal cytoreduction. Systematic pelvic lymph node dissection remains as a staging procedure only in cases of early-stage OC [34,71].

## 6. Suprarenal Lymph Nodes

The suprarenal lymph nodes lie cranial to the renal veins. They are divided into two groups. The first group is located on the left side of the aorta, inferior to the origin of the superior mesenteric artery and medial to the left adrenal gland. The second group is located between the IVC and the abdominal aorta, below the inferior surface of the right liver lobe and below the origin of the superior mesenteric artery. The boundaries of the dissection are as follows: cranial—inferior surface of the left liver lobe; caudal—left inferior suprarenal vein; left—medial portion of the left suprarenal gland; right—medial aspect of the IVC [72]. The importance of suprarenal lymph node metastases, previously considered a predictor of suboptimal cytoreduction, has been increasingly recognized as recent data reported improved feasibility of suprarenal lymphadenectomy using modified surgical techniques (instead of the conventional infrarenal paraaortic approach) [48,72,73,74]. Whereas most authors recommend the Kocher maneuver or its modification to gain access to the suprarenal lymph nodes [70,72,74], Daia et al. described a left lateral approach to the suprarenal nodes (modification of the Mattox maneuver) and concluded that the left approach provides a better exposure compared to the right one [73]. Surgeons should exercise caution when dissecting the adrenal glands, the suprarenal and renal vessels (anatomical variations of the renal vessels are common), the cisterna chyli and the lumbar lymphatic trunk. 

Komiyama et al. conducted the only study examining the metastatic pattern of suprarenal lymph nodes in AEOC [72]. The authors reported involvement of the suprarenal lymph nodes in 15% of cases. Nevertheless, the study looked at a small cohort of patients, including women with early-stage OC. In patients with advanced disease, the percentage rises to 36%. Notably, a preoperative CT scan revealed suprarenal disease in only one patient. The authors also noted that no patient developed isolated suprarenal lymph node metastases, as these were always associated with infrarenal involvement. Based on these findings, they suggested that patients with suprarenal lymph node involvement should be classified as stage IV because suprarenal disease may represent distant rather than regional dissemination [72,75]. Suprarenal lymph node dissection is managed by oncogynecologists [72].

## 7. Retrocrural Lymph Nodes

The retrocrural space (RCS) is a triangular area bounded antero-inferiorly by the two diaphragmatic crura, postero-superiorly by the mediastinal pleura and posteriorly by the thoraco-lumbar vertebra. The following anatomical structures are part of the RCS: vessels (aorta, azygos and hemiazygos veins), lymph nodes, fatty tissue, neural components (sympathetic trunk, splanchnic nerves), thoracic duct and cisterna chyli [76,77]. The RCS communicates with the retrocardiac space, the posterior mediastinum and the retroperitoneum. Moreover, this space provides communication between the thoracic paravertebral region and the celiac ganglion (Figure 14) [77]. Lymph nodes in this space are called “retrocrural” lymph nodes. These nodes communicate with the posterior mediastinal and paraaortic lymph nodes [78]. The progressive spread of lymph node metastases in a cranial direction can reach the retrocrural lymph nodes. A diameter of these nodes greater than 6 mm can be considered suspicious [76]. Im et al. investigated 67 OC patients at stage IV of the disease. The authors used 18F-FDG PET/CT to detect the presence of retrocrural lymph node metastases. The study found that 27 patients (40.3%) had metastases to the retrocrural lymph nodes [78]. Studies also reported that retrocrural lymph node metastases are associated with supradiaphragmatic lymph node involvement [77,78]. Retrocrural lymph nodes have recently been discovered for OC surgery [77,79]. However, there remains a need and potential for greater awareness of this space, relevant to surgical dissection and lymph node assessment. Therefore, preoperative evaluation of retrocrural lymph nodes through various imaging techniques can be of interest, although the diagnostic accuracy has to be determined. Relevant for the surgical practice is that iatrogenic injury of the cisterna chyli during exploration of the RCS can result in chylous leakage and chyloperitoneum [76,78]. The retrocrural space is not widely investigated in gynecologic oncology. Therefore, we recommend multidisciplinary surgical team management (general surgeons, oncogynecologist and thoracic surgeons) during retrocrural lymph node dissection.

Surgery of the retrocrural lymph nodes was first introduced in thoracic surgery and urologic surgery. Kern et al. reported on 211 patients with testicular malignant tumors, who underwent retrocrural dissection. The authors stated that their surgical approach has changed over the years. The study stated that the transabdominal/transdiaphragmatic approach at the time of midline retroperitoneal lymph node dissection was the most preferable option. Moreover, this approach was associated with fewer complications. The authors also reported that a multidisciplinary surgical team with urological and thoracic surgeons is beneficial for patients [80].

Sponholz et al. reported two approaches among germ cell cancer patients who underwent 50 retrocrural metastasectomies. The abdominal approach was performed in collaboration between a thoracic surgeon and a urologist. A bilateral transverse upper abdominal laparotomy was used. The liver and the right kidney were further mobilized, followed by the incision of the diaphragmatic crus. Mobilization of the spleen, left colonic flexure and left kidney was performed in cases of left-sided disease. The abdominal approach was preferable in cases of lower retrocrural, bilateral retrocrural and further abdominal metastases. The thoracic approach was performed in cases of upper retrocrural metastases. The authors concluded that the abdominal approach was associated with less tension at the spinal arteries and decreased risk of paresis compared to the thoracic approach [81]. 

Another urologic study described the dissection of the retrocrural lymph nodes by using a left thoraco-abdominal incision. The authors performed the Mattox maneuver in order to gain access to the retroperitoneal structures together with the retrocrural lymph nodes. The Mattox maneuver represents a left medial visceral rotation–medialization of the spleen, tail of the pancreas, left kidney and stomach. The initial incision starts at the line of Toldt from the sigmoid colon to the splenic flexure [82,83,84].

In gynecologic oncology, two studies analyzed the surgical access to the retrocrural lymph nodes. Song et al. performed the Kocher maneuver in order to access the retrocrural lymph nodes [85]. Another study described the resection of the gastrocolic and gastrosplenic ligaments with exposure of the pancreas in order to gain access to the retrocrural lymph nodes [86]. However, it seems that these two studies mixed the concept and exact anatomical location of the retrocrural lymph nodes, as they described a surgical dissection of the suprarenal lymph nodes [85,86]. More studies are needed to describe the safe and feasible surgical approach to the retrocrural lymph nodes in gynecologic oncology. The procedure should be performed by a multidisciplinary surgical team that includes oncogynecologists and thoracic surgeons.

## 8. Cardiophrenic Lymph Nodes

Cardiophrenic lymph nodes (CPLNs) are situated in the cardiophrenic region, which lies between the mediastinum, the base of the heart, the diaphragm and the chest wall. This region is composed of fatty tissue and lymph nodes that drain lymph from the diaphragm and abdominal organs. There are three groups of CPLNs. The anterior right group is bounded posteriorly by the pericardium and anteriorly by the xiphoid process. This group is further divided into two subgroups: the right subgroup (located to the right of the heart and in the inferior part of the mediastinum) and the left subgroup (located to the left of the sternum and anterior to the heart in the inferior part of the mediastinum). The middle group contains lymph nodes situated medial to the pericardium and lateral to the hilus of the lung at the region where the phrenic nerves pass. The posterior group of CPLNs is found near the esophagus, medial and posterior to the IVC at the level of the hiatus of the aorta. OC mainly affects the anterior group [87,88]. The optimal size cut-off for metastatic CPLNs is controversial and varies among authors. Some studies suggest a cut-off of 5 mm, whereas others propose a size greater than 7 or even 10 mm [87,88,89,90,91,92,93,94,95,96,97,98,99]. One study reported that preoperative PET/CT is more effective than CT scans in the detection of pathologic CPLNs [91]. Of note, most authors used a radiological cut-off of more than 5 mm [80,85,94,97]. Prader et al. found that 62% of patients with AEOC had suspicious CPLNs (>5 mm). The authors also observed that 84% of radiologically suspected CPLNs were metastatic [87]. Two studies using the same cut-off values found radiological CLPN adenopathy in 40.3% and 50% of the examined patients, respectively [92,94]. Cowan et al. emphasized that normal CPLNs usually measure less than 5 mm. Consequently, radiologically suspicious lymph nodes should have a diameter greater than 5 mm [97]. However, a cut-off of more than 7 mm used in other studies also showed good sensitivity (63%) and specificity (83%) [98]. The varying cut-offs used in different studies could contribute to an unequal incidence of radiologically detected CPLN adenopathy in AEOC patients. However, most studies reported a frequency of more than 50 percent, with malignancy being confirmed on the pathologic specimen in 45–95% of cases [87,88,89,90,91,92,93,94,95,96,97,98,99]. 

Most studies showed a strong correlation between radiographically enlarged CLPNs and more advanced peritoneal carcinomatosis in the abdomen [87,89,90,92]. Prader et al. showed that the right upper abdomen was the most affected region in patients with positive CLPNs [87]. Moreover, CLPN adenopathy is commonly associated with ascites and extra-abdominal disease [90]. Patients with radiologically positive CPLNs have decreased PFS and OS compared to women without radiological CPLN adenopathy. Luger et al. reported that CPLN adenopathy (>5 mm) and high CA-125 levels are independent prognostic factors for reduced PFS [92]. The therapeutic effect of removing metastatic CPLNs at the time of cytoreduction in patients with AOEC remains unclear [87,89,90,92,93]. However, suspicious CPLNs should not represent a contraindication to possible complete abdominal cytoreduction, as the latter remains the most favorable prognostic factor [92,99]. 

Surgical access to the CPLNs can be achieved through video-assisted thoracic surgery, trans-diaphragmatic or subxiphoid resection (substernal approach) [97,98,100]. The trans-diaphragmatic approach is particularly suitable for gynecologic cytoreductive procedures as it often does not require the use of a chest tube, and the patient’s position is not changed during surgery. The thoracic cavity is assessed by direct opening of the diaphragm to reach the CPLNs. Yoo et al. suggested that the trans-diaphragmatic approach might replace the video-assisted approach as it can be performed by oncogynecologists without significant complications [96]. This approach requires liver mobilization and diaphragmatic opening [90,96,97,98,99]. Therefore, some authors prefer the subxiphoid approach, where the pleural space is entered while the diaphragm is kept intact. Minig et al. recommended this approach for cases where the diaphragm remains intact after diaphragmatic stripping, to reduce postoperative complications related to entering the thoracic cavity (such as pleural effusion, pneumothorax, pneumonia) [100]. When dissecting the anterolateral cardiophrenic space, surgeons should exercise caution to avoid injury to the left phrenic nerve and the left pericardiophrenic artery and vein [15,100]. We do not suggest using monopolar energy devices during cardiophrenic lymph node dissection, as they could cause fatal heart arrhythmia. Oncogynecologists could perform cardiophrenic lymph node dissection in cases of trans-diaphragmatic surgery, whereas the thoracic approach requires thoracic surgeons [97,98,100]. The anatomy and metastases of cardiophrenic lymph nodes are shown in Figure 15 and Figure 16.

## 9. Inguinal Lymph Nodes Anatomy

Metastases to the inguinal lymph nodes (IGLNs) from OC are rare. Consequently, the inguinofemoral zone is often neglected during preoperative clinical and radiological examination. The incidence of metastases is approximately 3% to 5% among patients with AEOC [101,102,103,104,105,106,107], although metastatic enlargement of the IGLN as the first manifestation of AEOC has also been reported [108]. However, these data should be interpreted with caution, as some studies included autopsies of patients with advanced disease, whereas others focused on recurrent disease without detailed clinico-pathological information (e.g., possible association with pelvic or paraaortic metastases) [104,105,106]. Only a few reports described an interesting scenario with isolated IGLN metastases and OC [102,103,109,110]. IGLN adenopathy may be the first and only manifestation of lymphatic spread in OC [102,109]. 

The exact mechanism of OC metastases to IGLNs remains unclear. Lymphatic spread through the round ligament or hematogenous spread are thought to be the two main routes [101,102,103,104,105,106,107,108,109]. Kleppe et al. examined the drainage pathways of the ovaries of three female fetuses and one fresh cadaver, identifying two major and one minor drainage pathways. While the major pathways followed the course of the proper ovarian ligament (towards the obturator and internal iliac nodes) and the infundibulopelvic ligament (towards the paraaortic and paracaval lymph nodes), the third (inguinal) pathway drained the ovaries through the round ligament to the IGLNs. This pathway was not present bilaterally in all fetuses and tended to disappear during embryogenesis, while only a few lymph vessels may persist in a small percentage of patients. Therefore, the rarity of IGLN metastases in AEOC can be explained by the embryological development of the lymphatic system [111]. Nonetheless, it has been confirmed that peritoneal spread could also play a role in metastases to the IGLNs. Giri et al. documented peritoneal spread (“transcoelomic dissemination”) to the IGLNs along the inguinal hernia track in two patients with AEOC [101]. 

Palpation and imaging of the inguinofemoral region should be part of the diagnostic evaluation of every gyneco-oncological patient. If in doubt, a biopsy or advanced imaging (PET/CT) can be applied to confirm the diagnosis [101,108,112]. 

Some data questioned the classification of IGLN metastases as FIGO stage IV [102,113]. In a retrospective study, Nasioudis et al. compared the survival of patients with stage III and stage IV OC. The authors categorized the patients into four groups: group 1 (stage IV—IGLN metastases), group 2 (stage III—paraaortic/pelvic node metastases), group 3 (stage IV—metastases to distant nodes) and group 4 (stage IV—distant metastases). The study revealed that patients in groups 1 and 2 had similar outcomes. Additionally, patients in group 1 had a higher survival rate compared to those in groups 3 and 4 [113]. Another recent retrospective evaluation confirmed that the presence of IGLN metastases did not imply worse clinical outcomes compared to all stage III/IV patients, and R0 resection in AEOC patients with inguinal lymphadenopathy resulted in improved PFS [104]. 

The IGLNs are divided by the fascia lata into superficial and deep. The superficial IGLNs are located beneath the Scarpa fascia and separated into five groups, depending on the separation point of the great saphenous vein. The deep lymph nodes are located medial to the femoral vein and posterior to the fascia lata. The lymph node that makes the connection between the IGLNs and the iliac lymph nodes is called Cloquet’s node. This node is located anterosuperior to the femoral vein [114]. 

A skin incision (8–10 cm) is performed between the anterior superior iliac spine and the pubic tubercle. The incision is located parallel to and 2 cm below the inguinal ligament. Superficial IGLNs are removed between Scarpa’s fascia and the fascia lata. Deep IGLNs are removed after dissection of the cribriform fascia. It should be remembered that the anatomy of vital structures in the inguinal region, from lateral to medial, is as follows: femoral nerve, femoral artery, femoral vein, great saphenous vein. The great saphenous vein should be preserved in order to decrease the percentage of the most serious complication—lymphedema [114,115].

Metastatic inguinal lymph nodes from ovarian cancer are shown in Figure 17 and Figure 18.

## 10. Perioperative Scores Predicting Optimal Cytoreduction in Advanced Ovarian Cancer

Scores that predict the chance of achieving optimal cytoreduction in AEOC include combinations of imaging scores and intraoperative scores. 

Various radiological scores have been proposed, of which the most widely used remain the preoperative peritoneal cancer index score and the Suidan score [116,117,118,119,120,121]. The Suidan score is a predictive score based on three clinical (age ≥ 60 years, CA-125 ≥ 500 U/mL, ASA 3–4) and six imaging criteria (suprarenal or cardiophrenic lymph nodes larger than 1 cm; diffuse small bowel adhesions/thickening; and lesions more than 1 cm in the mesentery of the small bowel, in the root of the superior mesenteric artery, around the spleen and lesser sac area). The rate of suboptimal cytoreduction correlates with a higher score. Authors reported that a lesser sac lesion of more than 1 cm is a higher predictor for suboptimal cytoreduction. The authors of [116] did not mention retrocrural lymph nodes in their study. However, the limitations of imaging studies were associated with a low success rate when cross-validation datasets were used [116,117,118,119]. 

Jacquet and Sugarbaker developed the preoperative and intraoperative peritoneal cancer index (PCI) score for patients with mesothelioma and colorectal cancer (mucinous adenocarcinoma). Preoperative CT is used to define the degree of peritoneal carcinomatosis in the abdomen. The PCI includes nine abdominal regions. Additionally, four regions of the gastrointestinal tract are included—upper jejunum/lower jejunum and upper ileum/lower ileum. There is a scoring system from 0 to 3 for each of these regions (V0—absence of cancer, V1—tumor implants < 0.5, V2—tumor 0.5–5 cm, V3—tumor more than 5 cm). The score ranges from 0 to 39. A higher PCI is associated with suboptimal debulking, a higher rate of surgical complications and a worse prognosis [121]. One study found that patients with a PCI of more than 24 should be referred to neoadjuvant treatment [122]. However, the cut-off value reported in medical literature is between 10 and 20 [113,114,115]. A study by Climent et al. estimated that the highest sensitivity of PCI is observed with a cut-off of more than 20 [123]. Authors also concluded that the PCI is the best predictor of determining suboptimal cytoreduction for patients with peritoneal cancers [123]. The PCI score does not describe tumor implants in retroperitoneal lymph nodes, as it was validated for mesothelioma and mucinous rectal adenocarcinoma, although its higher index is often associated with lymph node involvement [124]. However, we believe that some areas of lymph node metastases could be neglected by using the PCI score as the only predictor for optimal cytoreduction. Moreover, the PCI score is not applicable in cases of OC with isolated lymph node dissemination [125,126].

The most widely used laparoscopic score in clinical practice is the Fagotti score. Fagotti et al. created a score based on laparoscopic predictive index value in order to estimate the probability of achieving optimal cytoreduction (residual tumor < 1 cm was the criterion during the study). Initially, the Fagotti score was based on the laparoscopic evaluation of seven parameters: peritoneal carcinosis, omental cake, diaphragmatic carcinosis, mesenteric retraction, stomach infiltration, liver metastases and bowel infiltration. Each parameter is valued as 0 (absence) or 2 (present). The total value varies between 0 and 14. Fagotti et al. stated that a score equal to or more than 8 is associated with suboptimal cytoreduction [127,128,129]. Later, with the advances in upper abdominal gynecologic oncology surgery, the Fagotti score was updated. A cut-off value of 10 was applied. Mesenteric retraction was excluded from the score [130,131,132]. Moreover, the score was also implemented for patients undergoing interval debulking surgery [129]. However, laparoscopic evaluation of cases of AEOC has some limitations—observation of the lesser sac, hilus of the spleen, gastrosplenic ligament, mesenteric root and retroperitoneal lymph nodes. Nevertheless, Fagotti et al. mentioned that the purpose of the score is to avoid an inappropriate lack of exploration, instead of unnecessary laparotomies [127,128]. Furthermore, disease involvement in areas unexplored by laparoscopy in the upper abdomen could be predicted regarding the dissemination of the disease [127,128]. There are other studies that try to estimate the role of laparoscopy in AEOC [133,134,135]. Di Donna et al. stated diagnostic laparoscopy should be integrated into the decision-making algorithm for patients with AEOC [120]. However, a Cochrane review concluded that laparoscopic evaluation is useful in identifying unresectable disease, whereas it has some limitations regarding the prediction of optimal cytoreduction [136]. The ENGOT (European Network of Gynaecological Oncology Trial) group reported that laparoscopy to access resectability was performed in only 25.4% of European centers, and among these, the laparoscopic score was not routinely used as a predictor model for treatment strategy [137]. The British Gynaecological Cancer Society guidelines do not recommend the routine use of laparoscopy to predict the resectability of the disease [138]. 

A laparoscopic assessment of the left upper abdomen is shown in Figure 19. 

The Eisenkop score intraoperatively estimates the extent of the OC disease in five regions in the abdomen—the right and left upper quadrants, pelvis, central abdomen and retroperitoneum. Each parameter is valued between 0 and 3. The total value varies between 0 and 15. Suprarenal, retrocrural and cardiophrenic lymph nodes are not mentioned in the scoring system [139]. However, suprarenal lymph nodes could be included as it was mentioned that retroperitoneal lymph nodes are removed below the crura of the diaphragm [139].

Aletti et al. created a score based on surgical complexity during surgery for AEOC. The authors included 12 surgical procedures, of which liver resection, splenectomy, diaphragm stripping/resection and large bowel resection were valuated with 2 points. Rectosigmoidectomy with anastomoses was 3 points. Other surgical complex procedures were valuated with 1 point—total hysterectomy with bilateral salpingo-oophorectomy, omentectomy, pelvic and paraaortic lymphadenectomy, stripping of the pelvic peritoneum, abdominal peritoneum stripping and small bowel resection. The authors estimated that a score of more than or equal to 8 is associated with high-complexity procedures [140]. However, celiac, suprarenal, retrocrural and cardiophrenic lymph node dissection was not described or included in the study [140].

Although the Aletti, Eisenkop, Sugarbaker and Suidan scores are from relatively old studies, their implementation in different scoring systems is still applicable [141,142,143]. However, a scoring system exploring the majority of neglected areas that may harbor residual disease and predicting preciously optimal cytoreduction in AEOC has not been validated yet. This unresolved issue is visible worldwide, as even artificial intelligence is used to predict optimal cytoreduction [144].

## 11. Preoperative Management for Patients Undergoing Surgery for Advanced Epithelial Ovarian Cancer

OC patients who undergo either a primary debulking surgery (PDS) or interval debulking surgery (IDS) should be preoperatively managed [145,146,147,148]. Recently, the European Society of Gynecological Oncology (ESGO) introduced guidelines for perioperative management of AEOC patients undergoing debulking surgery. The guidelines include preoperative management of patients through preoperative fluid replacements, bowel preparation, intraoperative prevention of hypothermia, etc. One of the aspects of the guidelines also highlighted the management of fragile patients [145]. The frailty index includes at least 30 parameters such as help with activities of daily living, lost weight in the last three months, muscle strength through hand grip test, the presence of diabetes, high blood pressure and nutritional screening. It is recommended to include at least 30 parameters in the frailty index. Studies show that a high frailty index is strongly associated with worse surgical outcomes and poorer OS [146,147]. Therefore, preoperative screening for frail patients is mandatory. Such patients are not candidates for primary optimal debulking surgery in AEOC [146,147]. Narasimhulu et al. estimated that a high risk of surgical morbidity and mortality during AEOC surgery is exhibited by a patient with one of the following three criteria—albumin levels < 35 g/dL, age ≥ 80, age 70–79 (and one of the following: ECOG status > 1, stage IV disease, complex surgery extended more than hysterectomy with bilateral salpingo-oophorectomy, omentectomy). Optimal cytoreduction was strongly associated with increased morbidity and mortality in patients in a high-risk group [148]. 

## 12. Primary Debulking Surgery Followed by Chemotherapy or Neoadjuvant Chemotherapy Followed by Interval Debulking Surgery

It is imperative to mention the difference in OS after a complete cytoreduction at the time of PDS or IDS for patients with AEOC. Chiva et al. performed a systematic review of the topic and observed a higher rate of complete cytoreduction in the IDS group compared to the PDS group. However, the authors reported longer median survival for women who underwent PDS compared to the group with IDS [149]. In 2010, Vergote et al. published EORTC 55971, the first prospective randomized trial comparing PDS followed by platinum-based chemotherapy versus platinum-based neoadjuvant chemotherapy (NACT) followed by IDS. The trial showed similar OS values between the two groups. However, patients in the IDS group had a higher rate of optimal debulking compared to the PDS group. Moreover, morbidity and mortality were lower in the IDS group [150]. The CHORUS trial was the second randomized, controlled trial that also investigated the difference in OS between PDS and IDS. The trial showed the non-inferiority of IDS and supported data from the EORTC 55971 trial [151,152]. The SCORPION trial was a randomized, single-institution trial that tried to show the superiority of NACT followed by IDS compared to PDS followed by chemotherapy for patients with AEOC. The trial failed to demonstrate the superiority of OS and PFS for patients in the IDS group compared to the PDS group. However, the trial also reported a lower level of postoperative complications in the IDS group. The rate of complete cytoreduction was significantly higher in the IDS group [153]. Despite the findings that NACT followed by IDS is not inferior to PDS followed by chemotherapy, IDS has not been widely accepted due to conflicting data [152]. Most guidelines for surgical treatment of AEOC still recommend PDS followed by chemotherapy for all patients who are surgical candidates for optimal cytoreduction [154,155]. It should also be noted that IDS of the anatomical neglected areas would be more challenging due to chemotherapy-induced fibrosis [150]. However, NACT followed by IDS is an option for patients who are poor surgical candidates and/or have inoperable disease [152].

Neglected anatomical areas and recommendations for their dissection during surgery for AEOC are summarized in Table 1. 

Table 1 shows that high PCI and ascites strongly correlate with transcoelomic dissemination of the neglected areas in the upper abdomen [5,7,13,33,35]. Extensive peritoneal carcinomatosis is a predictor of the peritoneal metastatic involvement of these areas. Studies suggest that dissection of OB could be omitted in cases of adhesions of Winslow’s foramen and the absence of enlarged paraaortic, mesenteric and suprarenal lymph nodes [5,7,13]. The liver area should be carefully evaluated in cases of peritoneal carcinomatosis at the Glisson capsule and PH [51,52,53,54,55,56,58]. The spleen should be always investigated for hilum metastases in the case of transcoelomic dissemination of the omental bursa and the presence of omental cake [5,67,68,69,70]. Suprarenal, retrocrural and cardiophrenic lymph nodes should be evaluated in cases of bulky paraaortic lymph nodes. Enlarged lymph nodes could be estimated preoperatively using imaging techniques. However, they should be also intraoperatively investigated by the surgeon, especially in cases of enlarged lymph nodes in other areas in the retroperitoneum [72,74,75,78,87,89]. As shown in the table, the available evidence in medical literature does not support the evaluation of some of the areas, and further studies are needed.

Finally, it is imperative to mention that optimal cytoreduction in AEOC patients is not always feasible despite the appropriate preoperative evaluation of the patients, favorable frailty index, surgical technique and evaluation of all of the neglected areas. Zivanovic et al. observed that optimal cytoreduction was performed in 81%, 63% and 39% of women with no, minimal and bulky upper abdominal disease, respectively. However, the authors reported a significant increase in optimal cytoreduction over the years, especially for patients with bulky upper abdominal disease [156]. It should be stressed that favorable treatment outcomes strongly correlate with hospital ovarian cancer surgical volume and the medical facilities where the patients are treated. High-volume hospitals provide the opportunity for women to receive care from surgeons with higher comprehensive surgery rates [157,158,159]. Therefore, in 2020, the ESGO published an updated version of quality indicators for AEOC surgery in order to offer the patients the specific skills, experience, organization and surgical care required for achieving optimal treatment [160].

## 13. Conclusions

OC is a malignant disease with different ways of tumor dissemination. Unlike other gynecological malignancies, AEOC often spreads through peritoneal and lymphatic dissemination to the upper abdomen. Surgeons should be familiar with the neglected anatomical areas that may contain residual disease in order to perform optimal cytoreduction whenever possible. These areas are commonly involved and should be rigorously evaluated in every patient with AEOC as they often preclude optimal cytoreduction. Conversely, leaving the neglected anatomical sites unexplored inadvertently leads to compromised treatment outcomes.

## Figures and Tables

**Figure 1 cancers-16-00285-f001:**
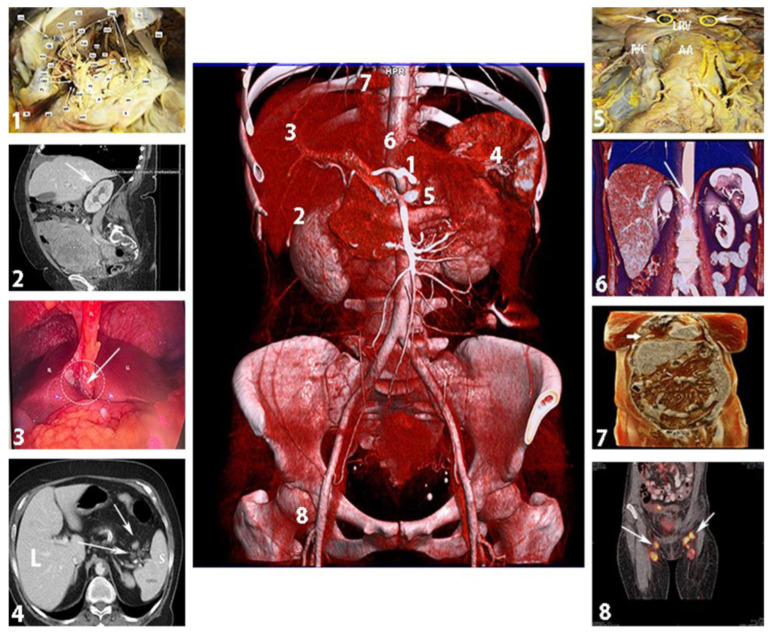
Neglected anatomical areas that may contain residual tumor in advanced ovarian cancer (authors’ own material). 1—omental bursa; 2—Morison’s pouch; 3—base of the round ligament and hepatic bridge; 4—splenic hilum; 5—suprarenal lymph nodes; 6—retrocrural lymph nodes; 7—cardiophrenic lymph nodes; 8—inguinal lymph nodes.

**Figure 2 cancers-16-00285-f002:**
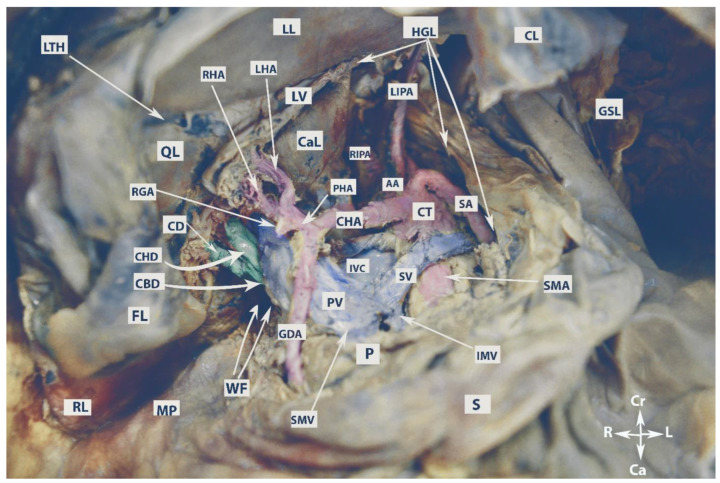
Anatomy of the supragastric omental bursa (embalmed cadaver, authors’ own material). LL—left lobe of the liver; RL—right lobe of the liver; LTH—ligamentum teres hepatis; QL—quadrate lobe; CaL—caudate lobe; FL—incised falciform ligament; CL—coronary ligament of the left liver lobe; GSL—gastrosplenic ligament; HGL—hepatogastric ligament; LV—ligamentum venosum; MP—Morison’s pouch; WF—Winslow’s foramen; CBD—common bile duct; CHD—common hepatic duct; CD—cystic duct; RGA—right gastric artery; LHA—left hepatic artery; RHA—right hepatic artery; GDA—gastroduodenal artery; PHA—proper hepatic artery; CHA—common hepatic artery; CT—celiac trunk; SA—splenic artery; LIPA—left inferior phrenic artery; RIPA—right inferior phrenic artery; AA—abdominal aorta; SMA—superior mesenteric artery; IVC—inferior vena cava; SV—splenic vein; PV—portal vein; SMV—superior mesenteric vein; IVM—inferior mesenteric vein; P—pancreas; S—stomach; Cr—cranial; Ca—caudal; L—left; R—right.

**Figure 3 cancers-16-00285-f003:**
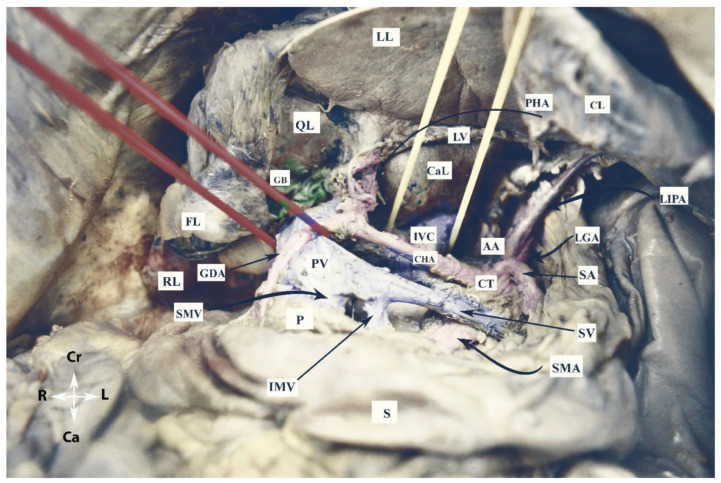
Vessels of the omental bursa (embalmed cadaver, authors’ own material). RL—right lobe of the liver; LL—left lobe of the liver; QL—quadrate lobe; CaL—caudate lobe; CL—coronary ligament of left lobe; FL—incised falciform ligament; LV—ligamentum venosum; S—stomach; P—pancreas; GB—gallbladder; CT—celiac trunk; AA—abdominal aorta; CHA—common hepatic artery; SA—splenic artery; GDA—gastroduodenal artery; LGA—left gastric artery; LIPA—left inferior phrenic artery; PV—portal vein; IVC—inferior vena cava; IMV—inferior mesenteric vein; SMV—superior mesenteric vein; PV—portal vein; SV—splenic vein; Cr—cranial; Ca—caudal; L—left; R—right.

**Figure 4 cancers-16-00285-f004:**
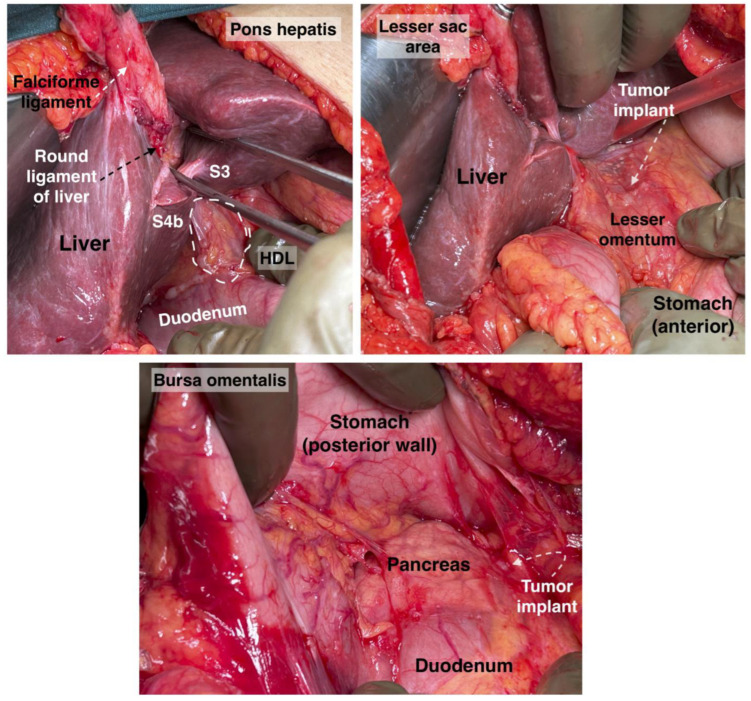
Pons hepatis, hepatoduodenal ligament, lesser sac area and bursa omentalis. Tumor implants will be found in those zones. S: segment, HDL: hepatoduodenal ligament. Surgical archive of author IS.

**Figure 5 cancers-16-00285-f005:**
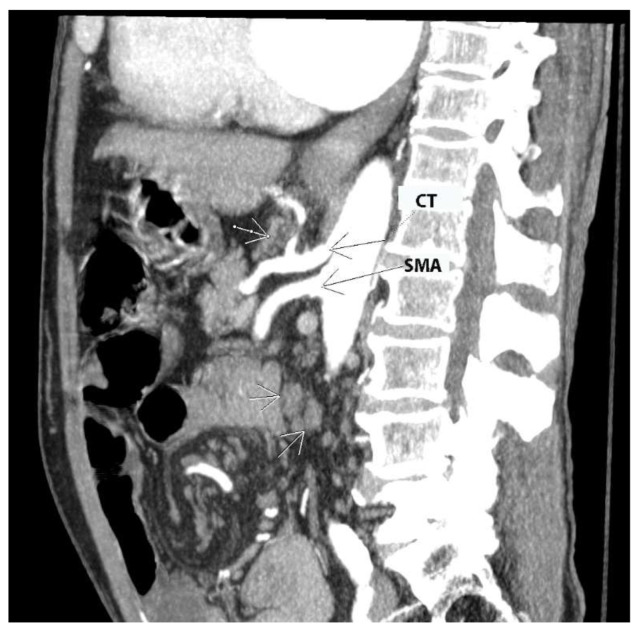
Contrast-enhanced CT in the sagittal plane (authors’ own material). Arrows point to pathologic lymph nodes in celiac and superior mesenteric stations. CT—celiac trunk; SMA—superior mesenteric artery.

**Figure 6 cancers-16-00285-f006:**
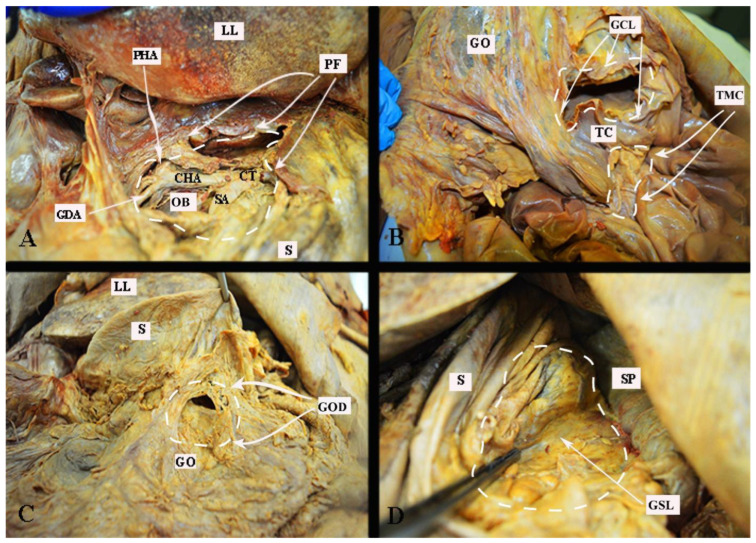
Surgical approaches to the omental bursa (embalmed cadaver, authors’ own material). (**A**) Dissection through the pars flaccida. (**B**) Trans-mesocolic approach and dissection of the gastrocolic ligament. (**C**) Direct dissection over the greater omentum just below the great curvature of the stomach. (**D**) Dissection of the gastrosplenic ligament. PF—pars flaccida; LL—left lobe of the liver; OB—omental bursa (supragastric part); GDA—gastroduodenal artery; CHA—common hepatic artery; CT—celiac trunk; SA—splenic artery; S—stomach; PHA—proper hepatic artery; GCL—gastrocolic ligament; TC—transverse colon; GO—greater omentum; TMC—trans-mesocolic dissection; GOD—greater omentum dissection just below its attachment to the great stomach curvature; SP—spleen; GSL—gastrosplenic ligament.

**Figure 7 cancers-16-00285-f007:**
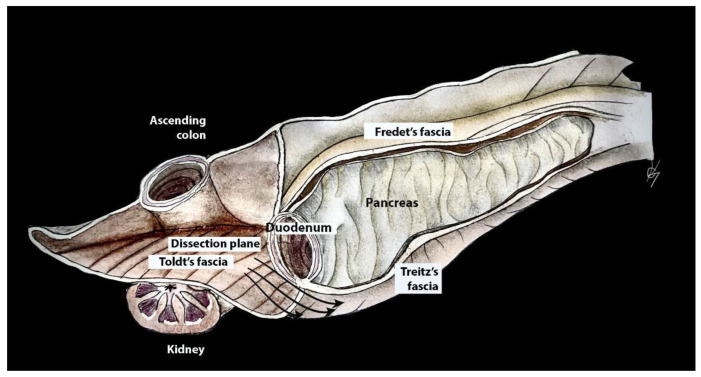
Dissection plane between the fusion fascia of Toldt and fusion fascia of Treitz during the Kocher maneuver (authors’ own material—modified from reference [48]).

**Figure 8 cancers-16-00285-f008:**
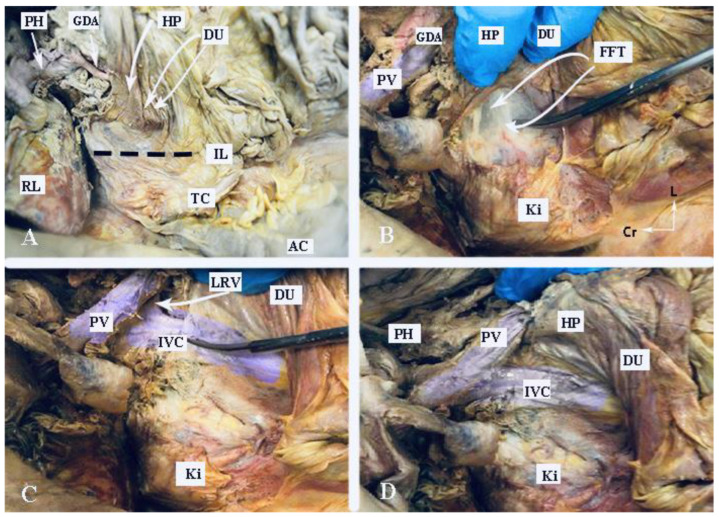
Kocher maneuver (embalmed cadaver, authors’ own material). (**A**) The duodenum and stomach are retracted medially. The incision line of the peritoneum is indicated by the interrupted black line. (**B**) Dissection of the fusion fascia of Treitz. (**C**) Mobilization of the duodenum and pancreatic head at the level of the left renal vein. The IVC is identified. (**D**) Anatomical structures after completion of the Kocher maneuver. PH—porta hepatis, GDA—gastroduodenal artery; HP—head of the pancreas; DU—duodenum; IL—incision line; RL—right lobe of the liver; TC—transverse colon; AC—ascending colon; PV—portal vein; Ki—kidney; FFT—fusion fascia of Treitz; LRV—left renal vein; Cr—cranial; L—left.

**Figure 9 cancers-16-00285-f009:**
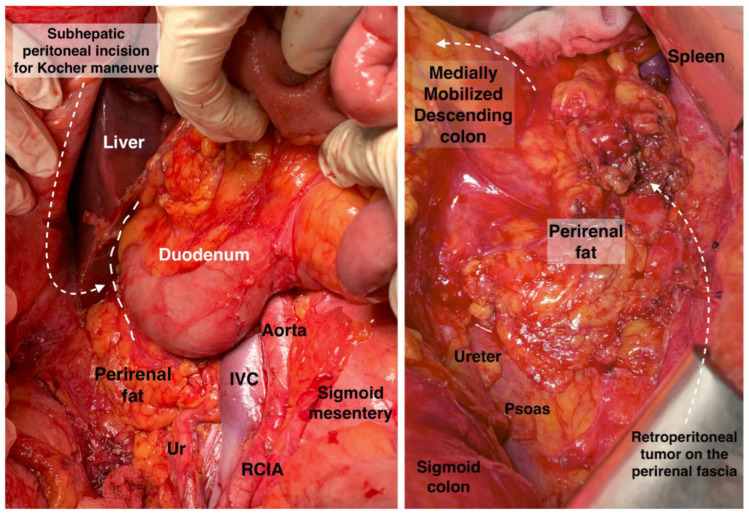
Anatomy for Kocher maneuver and retroperitoneal access on the (**right**) and (**left**) side. Ur: ureter, IVC: inferior vena cava, RCIA: right common iliac artery. Surgical archive of author IS.

**Figure 10 cancers-16-00285-f010:**
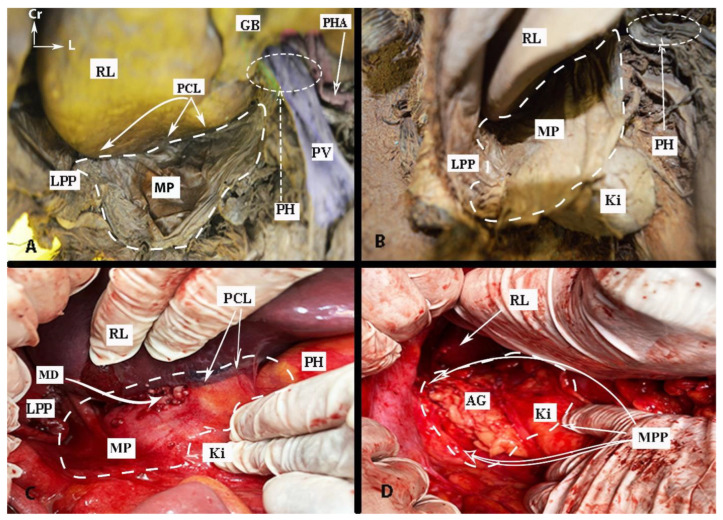
Anatomy of Morison’s pouch and tumor dissemination in ovarian cancer (authors’ own material). (**A**,**B**) Morison’s pouch anatomy. The boundaries of the pouch are pointed (embalmed cadaver). (**C**) Tumor dissemination of Morison pouch in advanced ovarian cancer (open surgery). (**D**) Final intraoperative view after peritonectomy of Morison’s pouch. The posterior layer of the coronary ligament on the right lobe and right triangular ligament were dissected. RL—right liver lobe; GB—gallbladder; PCL—posterior layer of the coronary ligament of the right liver lobe; LPP—lateral pelvic peritoneum; PH—porta hepatis; PV—portal vein; PHA—proper hepatic artery; MP—Morison’s pouch; Ki—kidney; MD—metastatic disease; AG—right adrenal gland; MPP—Morison’s pouch peritoneum dissection; Cr—cranial; L—left.

**Figure 11 cancers-16-00285-f011:**
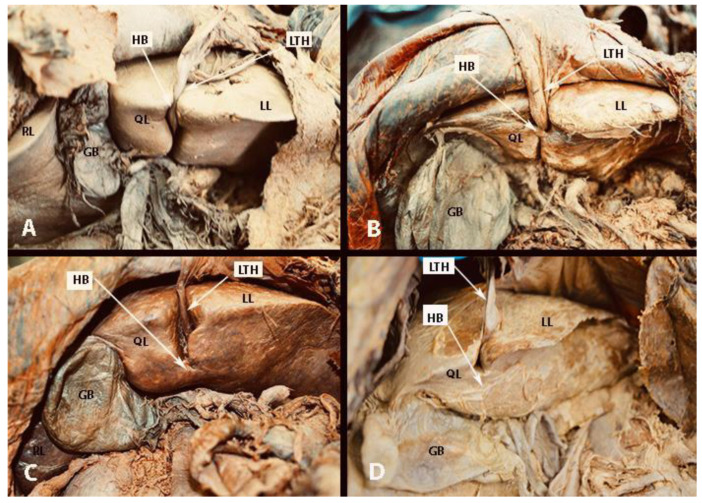
Types of hepatic bridge according to the Sugarbaker classification (embalmed cadavers, authors’ own material). (**A**) Type 0: no hepatic bridge, where the base of the round ligament is visible. (**B**) Type I: less than one-third of the umbilical fissure is covered by liver parenchyma; (**C**) Type II: the hepatic bridge covers up to two-thirds of the fissure; (**D**) Type III: more than two-thirds is covered by the liver parenchyma. LL—left liver lobe; RL—right liver lobe; QL—quadrate lobe; GB—gallbladder; LTH—ligamentum teres hepatis; HB—hepatic bridge.

**Figure 12 cancers-16-00285-f012:**
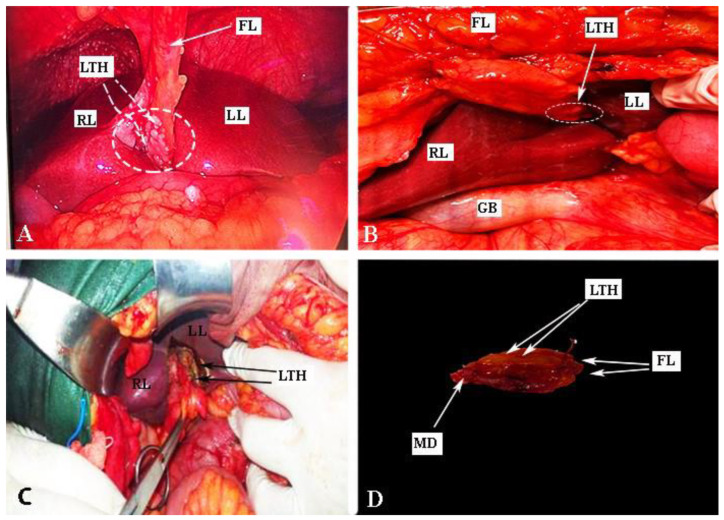
Metastatic spread to the base of the ligamentum teres hepatis (authors’ own material). (**A**,**B**) Metastases at the base of the round ligament ((**A**) laparoscopic surgery, (**B**) open surgery). (**C**) Transection of ligamentum teres hepatis (open surgery). (**D**) Postoperative specimen of the ligament. LTH—ligamentum teres hepatis; FL—falciform ligament; LL—left liver lobe, RL—right liver lobe; GB—gallbladder; MD—metastatic disease.

**Figure 13 cancers-16-00285-f013:**
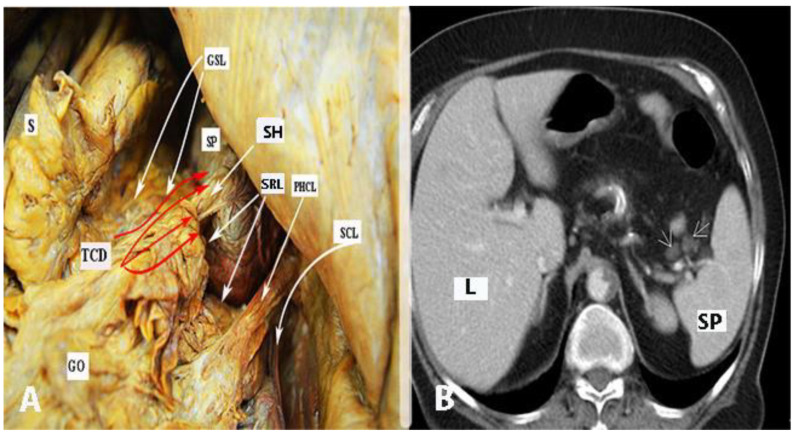
Dissemination of advanced ovarian cancer to the spleen (authors’ own material). (**A**) Spleen and its connection with the greater omentum. The red arrows show the extension of the spread through the greater omentum into the splenic hilum. (**B**) Contrast-enhanced CT of the upper abdomen in a transverse plane. Spleen is shown. The arrows point to pathological lymph nodes at the splenic hilum. S—stomach; SP—spleen; L—liver; TCD—tumor cell dissemination; GO—greater omentum; GSL—gastrosplenic ligament; SRL—splenorenal ligament; SH—splenic hilum; PHCL—phrenicocolic ligament; SCL—splenocolic ligament; SH—splenic hilum.

**Figure 14 cancers-16-00285-f014:**
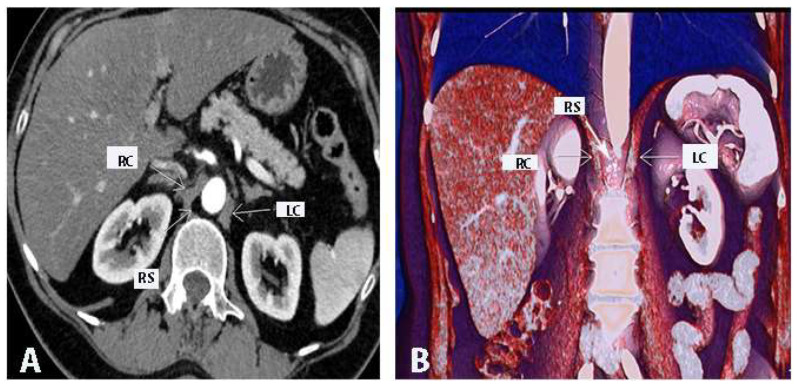
Retrocrural space (authors’ own material). (**A**) RCS: contrast-enhanced abdominal CT in the transverse plane. The arrows are pointing to the left and right diaphragmatic crura (authors’ own material). (**B**) RCS: 3D volume-rendered abdominal CT in the coronal plane. The arrows are pointing to the left and right diaphragmatic crura and the RCS. RC—right diaphragmatic crus; LC—left diaphragmatic crus; RS—retrocrural space.

**Figure 15 cancers-16-00285-f015:**
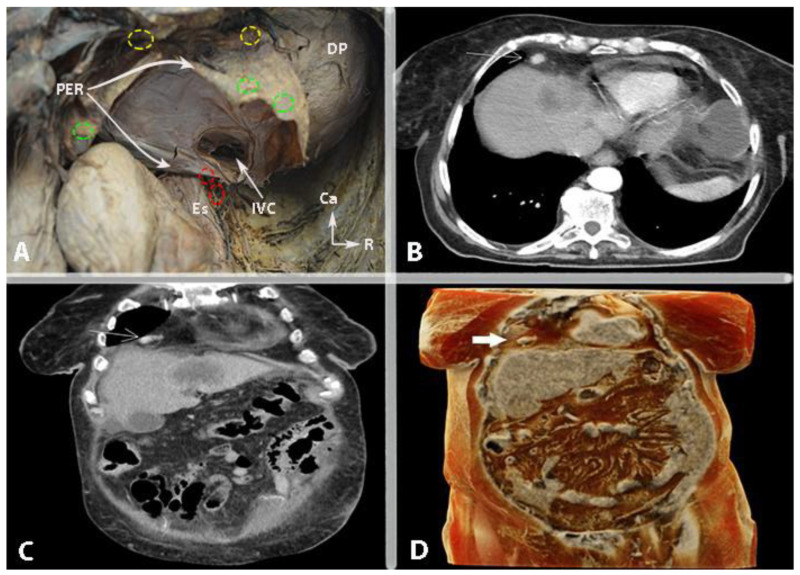
Cardiophrenic lymph nodes (authors’ own material). (**A**) Anatomical location of the anterior cardiophrenic lymph nodes (yellow circles—anterior group location; green circles—middle group location; red circles—posterior group location). (**B**) Contrast-enhanced CT in the axial plane. The arrow points to the metastatic cardiophrenic lymph node. (**C**) Contrast-enhanced CT in the coronary plane. The arrow indicates the metastatic cardiophrenic lymph node. (**D**) Three-dimensional CT volume-rendered image. The arrow points to the metastatic cardiophrenic lymph node. PER—pericardium; Es—esophagus; IVC—inferior vena cava; DP—diaphragmatic part of parietal pleura; Ca—caudal, R—right.

**Figure 16 cancers-16-00285-f016:**
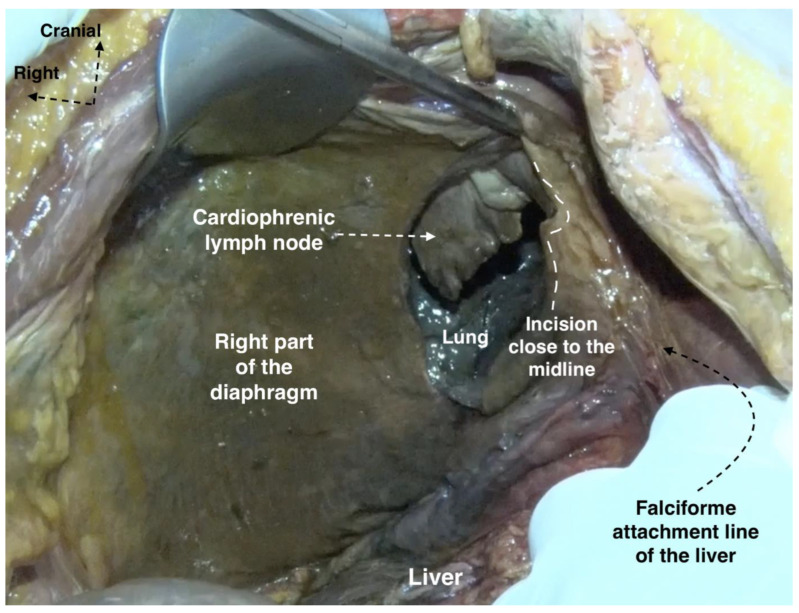
Cadaveric anatomy for cardiophrenic lymph node management (cadaveric dissection archive of author IS).

**Figure 17 cancers-16-00285-f017:**
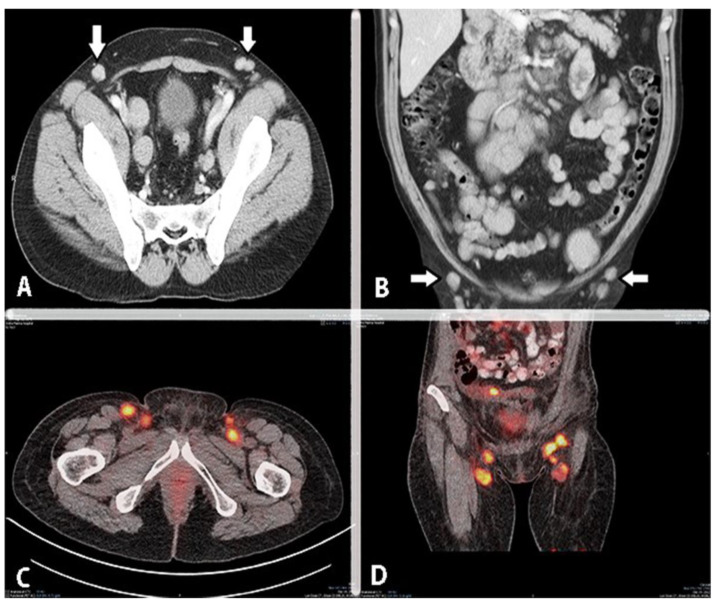
Imaging of metastatic inguinal lymph nodes in ovarian cancer (authors’ own material). (**A**) Contrast-enhanced axial CT image of the pelvis. The arrows point to pathologic inguinal lymph nodes. (**B**) Contrast-enhanced coronal CT image of the pelvis. Arrows point to pathologic inguinal lymph nodes. (**C**,**D**) PET/CT—inguinal lymph node metastases.

**Figure 18 cancers-16-00285-f018:**
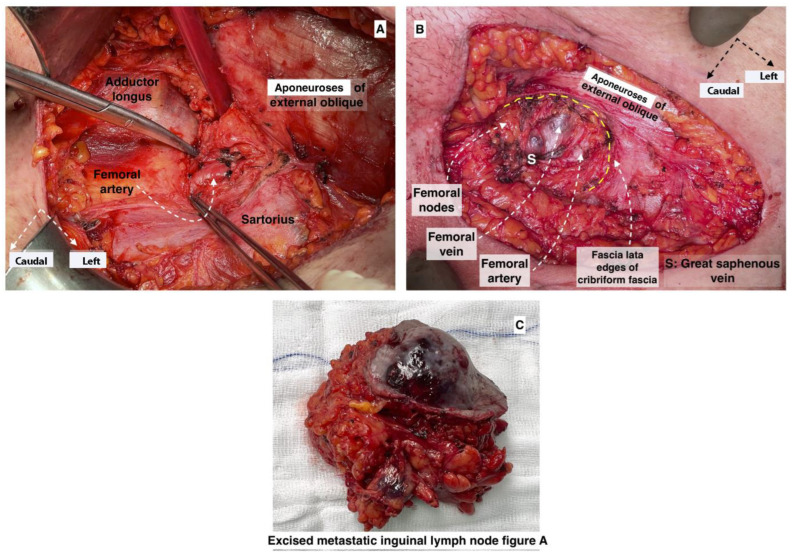
Inguinofemoral anatomy and metastatic bulky lymph nodes (surgical archive of author IS). (**A**) dissection of metastatic inguinal lymph node at the left inguinal region. (**B**) inguinofemoral anatomy after dissection. (**C**) excised metastatic inguinal lymph node.

**Figure 19 cancers-16-00285-f019:**
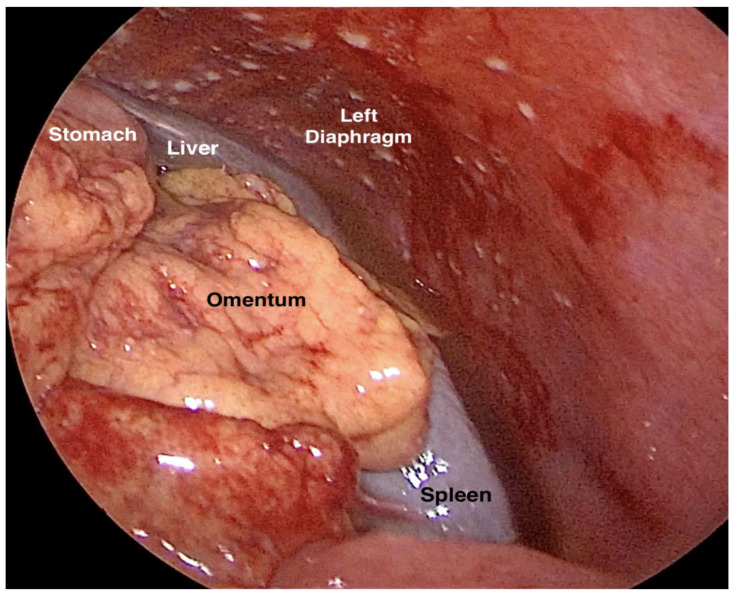
Laparoscopic evaluation of the feasibility of primary cytoreduction, control of left upper abdomen (laparoscopy performed by author IS). Peritoneal diaphragmatic carcinomatosis and absence of metastases to the greater omentum and spleen are seen.

**Table 1 cancers-16-00285-t001:** Recommendations concerning neglected anatomical areas of ovarian cancer. PCI—peritoneal cancer index; PH—porta hepatis; HDL—hepatoduodenal ligament.

Anatomical Area	Evaluation—Recommended	Evaluation—Not Supported by Available Evidence
*Omental bursa—transcoelomic dissemination [5,7,13,30,32,33,35]*	PCI ≥ 17, peritoneal carcinomatosis in the upper abdomen, ascites, Morison’s pouch and diaphragmatic peritoneum dissemination	Adhesion in Winslow’s foramen
*Omental bursa—lymph node dissemination (celiac, portal, triad) [5,7,13,30,32,33,35]*	Enlarged paraaortic, mesenteric and suprarenal lymph nodes	Normal paraaortic, mesenteric and suprarenal lymph nodes
	Suspicious metastatic mediastinal lymph nodes as judged using imaging techniques	
*Morison’s pouch [51,52,53,54,55]*	High PCI, ascites, peritoneal carcinomatosis, diaphragmatic dissemination, transcoelomic dissemination to the HDL	Isolated lymph node dissemination in the upper abdomen
*Base of the round ligament of the liver [56,58,60]*	PCI ≥ 10, ascites, peritoneal carcinomatosis at Glisson capsule and PH	Absence of peritoneal carcinomatosis at Glisson capsule and PH
*The hepatic bridge [56,58,60]*	Transcoelomic involvement of the round ligament, PCI ≥ 10	Absence of peritoneal carcinomatosis at Glisson capsule and PH
*Hilum of the spleen [5,66,67,68,69,70]*	Omental cake (greater omentum), transcoelomic OB involvement	Isolated lymph node dissemination in the upper abdomen
*Suprarenal lymph nodes [72,74,75]*	Enlarged infrarenal paraaortic lymph nodes	Normal infrarenal lymph nodes
*Retrocrural lymph nodes [77,78]*	Enlarged paraaortic, suprarenal, celiac and cardiophrenic lymph nodes	Normal paraaortic, suprarenal, celiac and cardiophrenic lymph nodes
*Cardiophrenic lymph nodes [87,89,90,96]*	Transcoelomic and lymphatic dissemination in the right upper abdomen, ascites, extra-abdominal disease	Not applicable
*Inguinal lymph nodes [101,103,109,110]*	Every patient. Particular attention should be paid for patients with inguinal hernia	Not applicable

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
