# Peer review of "Neglected Anatomical Areas in Ovarian Cancer: Significance for Optimal Debulking Surgery"

_cancers, 2024, doi:10.3390/cancers16020285_

Round 1
Reviewer 1 Report
Comments and Suggestions for Authors
Ovarian cancer is one of the most lethal malignancy in Gynecology, therefore an adequate surgical approach and its improvement are essential to increase the prognosis of this pathology. For this purpose, a detailed anatomical knowledge is the key to a successful intervention. At the same time, imaging techniques play an important role in preoperative diagnosis, in the future, more accurate imaging methods for determing metastases will improve the management of ovarian cancer.
The manuscript is clear, relevant to the field and presented in a well-structured manner. The background of the subject is well shaped. The manuscript is scientifically interesting and original. English language is correctly used.
The references are relevant and mostly from recent publications and do not include an excessive number of self-citations.
The figuers and table are appropriate, easy to understand and clearly explained.
The conclusions are drawn coherent, underlying the importance of knowing the detailed anatomy by surgeons and the need to explore the hidden areas in order to perform an optimal cytoreduction.
Author Response
Dear Reviewer,
Thank you very much for approving and highlighting our article!
Ovarian cancer is one of the most lethal malignancy in Gynecology, therefore an adequate surgical approach and its improvement are essential to increase the prognosis of this pathology. For this purpose, a detailed anatomical knowledge is the key to a successful intervention. At the same time, imaging techniques play an important role in preoperative diagnosis, in the future, more accurate imaging methods for determing metastases will improve the management of ovarian cancer.
The manuscript is clear, relevant to the field and presented in a well-structured manner. The background of the subject is well shaped. The manuscript is scientifically interesting and original. English language is correctly used.
The references are relevant and mostly from recent publications and do not include an excessive number of self-citations.
The figures and table are appropriate, easy to understand and clearly explained.
The conclusions are drawn coherent, underlying the importance of knowing the detailed anatomy by surgeons and the need to explore the hidden areas in order to perform an optimal cytoreduction.
We are deeply grateful for your comprehensive review.
Reviewer 2 Report
Comments and Suggestions for Authors
As the authors point out, the goal of optimal resection in ovarian cancer surgery is often based on the surgeon's gross assessment. This review paper focuses on the possibility that the presence of anatomically overlooked hidden tumors may preclude truly complete resection, and provides a detailed anatomical description of the sites that are likely to leave "hidden lesions" and the surgical procedures essential for dissection of these sites. The content of this paper is a valid summary of the anatomical findings, but falls short of the title "Significance of optimal cytoreduction in ovarian cancer".
1. Authors mainly mention anatomical features in each section of the manuscript. Can they describe how often metastases are found at each site and what specific procedures are needed to achieve complete removal?
2. Authors could mention whether these procedures should be performed by a surgeon or by a gynecologic oncologist? The authors should also add a discussion of how far the initial surgery should go and the difference in prognosis between PDS and IDS.
3. Table 1 describes the conditions for which an evaluation should be recommended. Authors need to discuss this content in more detail. The final conclusion, compromised treatment outcomes, is rarely mentioned.
Author Response
"Please see the attachment"

Reviewer 3 Report
Comments and Suggestions for Authors
The last name of the last author is Yordanov. In the email address it is written with a j (jordanov). Is this correct?
Ik the first alinea you state that advanced cancer spreads in most advanced stages to the upper abdomen. That is general speaking not true: stage 4 spreads beyond the abdominal cavity.
Reference number 1 and 83 are the same
Author Response
Dear Reviewer,
We are deeply grateful for your comprehensive review. Thank you for your insightful and constructive review of our paper on the anatomical challenges in ovarian cancer debulking surgery.
We incorporated the recommended changes. All incorporated changes are highlighted by using the Track and Changes in Word.
The last name of the last author is Yordanov. In the email address it is written with a j (jordanov).
Is this correct?
Author’s Reply:
The name on the manuscript is correct! It is Yordanov. In the email is different, but also correct!
Ik the first alinea you state that advanced cancer spreads in most advanced stages to the upper
abdomen. That is general speaking not true: stage 4 spreads beyond the abdominal cavity.
Author’s Reply: We agree with the reviewer! The next text was incorporated:
The main routes of spread include peritoneal and lymphatic dissemination with the upper abdomen
being commonly affected in advanced stages, which, in turn, increases the rate of lymph node and
peritoneal metastatic involvement and decreases the chance for complete cytoreduction [1].
Reference number 1 and 83 are the same
Author’s Reply: Thank you for the correction! Reference 1 has been changed. Another reference has
been incorporated instead.
1. Sehouli J, Senyuva F, Fotopoulou C, et al. Intra-abdominal tumor dissemination pattern and
surgical outcome in 214 patients with primary ovarian cancer. J Surg Oncol. 2009;99(7):424-427.
doi:10.1002/jso.21288
We are grateful for your valuable time and effort in reviewing our manuscript.
Based on your useful and scientific comments, we believe our manuscript has been improved to a higher level.
Reviewer 4 Report
Comments and Suggestions for Authors
The authors present an extensive thoroughly material on the anatomy and surgical approach of "hidden areas" for optimal cytoreduction in ovarian cancer. The title is adequately chosen, being consistent with the content presented. The article is well structured in chapters and subchapters presenting a comprehensive anatomy report, surgical approaches for every hidden area described and an up to date literature review at the end of each chapter, supported by important references. A suggestion could be to place the literature data from studies together in a discussion chapter at the end of the paper. However, this manner to place discussion at the end of each chapter is acceptable. The article raises awareness about these "hidden areas" thoroughly presented, their surgical approach definitely having clinical importance in the prognosis of ovarian cancer. The conclusion is relevant to the content presented. The paper is well written, the English language is acceptable, especially for non-native English speakers, but some expressions need rephrasing in a more academic style. For example, in the first sentence in the introduction "the most deadly gynecologic malignancy" sounds improper.
Comments on the Quality of English LanguageMinor editing, mostly rephrasing for some expressions.
Author Response
Dear Reviewer,
Thank you for your insightful and constructive review of our paper on the anatomical challenges in ovarian cancer debulking surgery. We incorporated the recommended changes. All incorporated
changes are highlighted by using the Track and Changes in Word.
The authors present an extensive thoroughly material on the anatomy and surgical approach of "hidden areas"; for optimal cytoreduction in ovarian cancer. The title is adequately chosen, being consistent with the content presented. The article is well structured in chapters and subchapters presenting a comprehensive anatomy report, surgical approaches for every hidden area described and an up to date literature review at the end of each chapter, supported by important references.
A suggestion could be to place the literature data from studies together in a discussion chapter at the end of the paper. However, this manner to place discussion at the end of each chapter is acceptable.
Author’s Reply: Initially we though to summarize literature data from studies together in a discussion chapter, but realized it would be hard for readers to preciously follow their thoughts! Therefore, we decided that it will be better to unite anatomy, metastases, and surgical approaches in a discussion at the end of each chapter.
The article raises awareness about these "hidden areas" thoroughly presented, their surgical approach definitely having clinical importance in the prognosis of ovarian cancer. The conclusion is relevant to the content presented. The paper is well written, the English language is acceptable, especially for non-native English speakers, but some expressions need rephrasing in a more academic style. For example, in the first sentence in the introduction "the most deadly gynecologic malignancy"; sounds improper.
Author’s Reply: The first sentence was changed! The whole manuscript was carefully revised again!
The next sentence was inserted:
Ovarian cancer (OC) is a rare disease with specific tumor biology and clinical behavior. Therefore, OC represents one of the major causes of lethality from cancer among women in developed countries [1].
We are grateful for your valuable time and effort in reviewing our manuscript.
Based on your useful and scientific comments, we believe our manuscript has been improved to a higher level.
Reviewer 5 Report
Comments and Suggestions for Authors
This is a well written and very thorough anatomic description of epithelial ovarian cancer spread to areas that usually prevent optimal debulking. However, the review is too long and not very focused.
The following changes are suggested:
1. The anatomical review in pages 3-6 should be removed. If the reader wants to know the exact anatomy it can be found in every anatomy book
2. I think that the review should focus on upper abdominal disease which indeed is the most neglected during debulking. I suggest removing pages 21-24 with the descriptions of extra-peritoneal lymph nodes. Axillary, inguinal and cardiophrenic lymph nodes are not hidden and do not pose major surgical difficulty.
3. The authors describe surgical methods to reach problematic metastatic sites. They do not describe a method for the retrocrural lymph node removal – please remove this section or offer surgical description as for the other sites.
4. The name of the review should be changed. The areas are not hidden. We see them on CT and PET CT prior to the operation. We see and palpate them during the operation, however since they are difficult to resect these areas are usually not approached. I think this is the importance of the present review. Please correct/change "hidden" also in the review itself
5. The review should be more focused on the difficulty to achieve optimal debulking. I think the authors should start with section X describing the different scoring methods and then aim to the problematic usually not debulkable areas and how to reach them.
6. Table 1 should be corrected according to the proposed changes.
7. Several Figures should be removed according to the proposed changes.
Author Response
"Please see the attachment"
